# Emission enhancement of erbium in a reverse nanofocusing waveguide

Nicholas A. Güsken ®[1,2] ✉, Ming Fu ®[1], Maximilian Zapf ®[3], Michael P. Nielsen ®[1,4], Paul Dichtl[1], Robert Röder[3], Alex S. Clark[1,5], Stefan A. Maier[1,6], Carsten Ronning ®[3] & Rupert F. Oulton ®[1] ✉

Since Purcell's seminal report 75 years ago, electromagnetic resonators have been used to control light-matter interactions to make brighter radiation sources and unleash unprecedented control over quantum states of light and matter. Indeed, optical resonators such as microcavities and plasmonic antennas offer excellent control but only over a limited spectral range. Strategies to mutually tune and match emission and resonator frequency are often required, which is intricate and precludes the possibility of enhancing multiple transitions simultaneously. In this letter, we report a strong radiative emission rate enhancement of $Er^{3+}$-ions across the telecommunications C-band in a single plasmonic waveguide based on the Purcell effect. Our gap waveguide uses a reverse nanofocusing approach to efficiently enhance, extract and guide emission from the nanoscale to a photonic waveguide while keeping plasmonic losses at a minimum. Remarkably, the large and broadband Purcell enhancement allows us to resolve Stark-split electric dipole transitions, which are typically only observed under cryogenic conditions. Simultaneous radiative emission enhancement of multiple quantum states is of great interest for photonic quantum networks and on-chip data communications.

The influence of a local electromagnetic environment on light-matter interaction offers a route to realize faster and brighter light sources with associated control over the quantum states of light and matter. Mastering this control is especially important in sensing and quantum technologies[1] when working with low photon numbers and single emitters such as isolated molecules[2–5], quantum dots[6] and atoms[7,8], all of which inherently interact particularly weakly with light. In the absence of a structured electromagnetic environment, luminescence is distributed across a continuum of states, yielding undirected radiation that can be difficult to harness. Consequently, over the past 20 years research has focused on controlling luminescence by directing it into electromagnetic cavity modes using the Purcell effect[9].

Optical cavities localize light in both frequency, quantified by the cavity quality factor, $Q$, and space, quantified by a mode volume, $V_m$, to accelerate luminescence by the Purcell factor, $F_p = 3\lambda^3 Q/(4\pi^2 n^3 V_m)$, where $\lambda$ is the optical wavelength and $n$ is the refractive index. This has proven to be incredibly effective, with recent works demonstrating exquisite control over individual atomic states[7,10,11]. However, this comes at the price of requiring innovative ways to tune and match both cavity and emitter frequency as well as the restriction of access to a single electronic transition at a time. Moreover, the emission rate of photons from a photonic cavity, $\gamma_{cav}$, scales as $\gamma_{cav} \propto Q^{-1}$. Hence, a cavity with $Q \sim 10^5$ for example, can only collect photons at GHz rate from a single emitter[12]. In fact, such cavities commonly require low-temperature operation to assure sufficiently long coherence times of the emitter placed within them[13]. A promising alternative is to couple luminescence into a single highly confined optical waveguide mode[14–16]. Now the mode's confinement area, $A_m$ and group velocity,

[1]Department of Physics, Imperial College London, Prince Consort Road, London SW7 2AZ, UK. [2]Department of Materials Science and Engineering, Stanford University, Stanford, CA 94305, USA. [3]Friedrich-Schiller-Universität Jena, Max-Wien-Platz 1, 07743 Jena, Germany. [4]School of Photovoltaics and Renewable Energy Engineering, UNSW Sydney, Kensington, NSW 2052, Australia. [5]Quantum Engineering Technology Labs, University of Bristol, Bristol BS8 1UB, UK. [6]Monash University School of Physics and Astronomy, Clayton, VIC 3800, Australia. ✉e-mail: ngusken@stanford.edu; r.oulton@imperial.ac.uk

$v_g$, determine the Purcell factor, $F_p = 3c\lambda^2/(4\pi n^2 v_g A_m)$[12,17], while the continuum of optical states provides tuning-free enhancement across a broad frequency range[18,19]. This in principle should provide simultaneous access to multiple electronic transitions at distinct energies[20]. Thus, plasmonic waveguides are especially promising as they offer large bandwidth sub-wavelength confinement $\left(A_m \ll (\lambda/2n)^2\right)$ without excessive dispersion[21–23], potentially at THz photon emission rates[13]. Furthermore, significantly stronger Purcell enhancements are achievable in plasmonic systems compared to their photonic counterparts[12,14,15,24–26], where increases in confinement have been driven by improvements in nanofabrication technology.

One critical attribute of strongly mode-confining plasmonic systems, which has so far remained elusive, is the efficient and guided extraction of enhanced emitter luminescence. In order to provide a technologically viable and scalable platform, it is key to not only strongly enhance emission but also subsequently feed it into a photonic network while keeping the inherent plasmonic losses at a minimum. In this letter, we demonstrate both the enhancement and efficient guided mode extraction of dipole luminescence[27] using reverse nanofocusing[21,28] from a nanoplasmonic waveguide coupled to a buried photonic waveguide[29–31]. This is realized by stacking a tapered plasmonic waveguide on top of an buried oxide photonic slab waveguide. While this keeps fabrication to a minimum, coupling to buried ridge waveguides is equally possible. Our hybrid coupling approach eases device fabrication requirements and allows seamless integration of plasmonic waveguides with existing Si-photonic integrated circuit technology. We demonstrate this using technologically relevant $Er^{3+}$-ions. Rare-earth ions recently re-gained attention for their prospective applications in quantum photonic networks[7,32–35] and optical communications[36]. Such emitters are highly desirable[11] as they offer robust, uniform and reproducible quantum states with emission bands that are suitable for quantum photonic networks at telecommunication wavelengths. Unfortunately, erbium's inherently long lifetimes (~10 ms)[37] have hindered practical exploitation for nanophotonics: a single $Er^{3+}$-ion offers only ≈100 photons per second with ideal collection. Notably, our reverse nanofocusing device gives rise to 300-fold shorter $Er^{3+}$-ion lifetimes accompanied by an enhancement of >338 in luminescence efficiency when compared to non-plasmonic control

devices. Efficient collection of the enhanced luminescence is achieved by exploiting photonic to plasmonic nanofocusing methods, but in reverse. The extraordinary enhancements of this system are further underpinned by the observation of multiple enhanced Stark-split electric dipole transitions across the $Er^{3+}$ telecommunications emission band. This demonstrates the capability to strongly dress multiple atomic transitions simultaneously using a one-dimensional mode continuum at room temperature. The plasmonic waveguide presented, provides a Purcell factor of >338 over a large bandwidth. In comparison, waveguide-coupled photonic cavities have been shown to provide Purcell factors of >700, but over sub-nm bandwidths[38]. Although slow-light photonic crystal waveguides can extend the bandwidth to a few nms, Purcell factors (~10)[39] are set by their delay-bandwidth product[40]. Finally, photonic slot waveguides have been shown to provide modest Purcell enhancements in the range of 8–35[41].

The non-resonant hybrid waveguide device combines several unique characteristics: i) broadband emission enhancement, ii) rapid, efficient and guided photon extraction from quantum emitters, iii) facile and compact integration into photonic networks and iv) the possibility to simultaneously address emitters optically and electrically to e.g. tune Stark split states by external electrical fields[42]. The device – when combined with a hybrid integration approach[43] – provides a novel, scalable route with the potential to enable next-generation on-demand[44] single photon sources for quantum photonic integrated circuits (qPICs)[45].

Here, photonic quantum emitter-coupled systems relying on high-Q cavities, give rise to large emission enhancement, good modal coupling and guiding. However, storing photons for many cycles fundamentally limits emission rate and bandwidth. Plasmonic antenna's such as plasmonic nanocubes[3,46] enable emission enhancements of >10³ and large cavity emission rates. However, the guided extraction of light in an antenna-coupled quantum emitter system is difficult, hindering integration into PICs. Meanwhile, the bandwidth is limited by the antenna resonance and the spatial overlap of excitation and emission introduces challenges for excitation beam filtering. The integrated waveguide presented here in contrast, is non-resonant and hence inherently broadband, not bandwidth-delay product limited, separates excitation and emission spatially and provides a Purcell

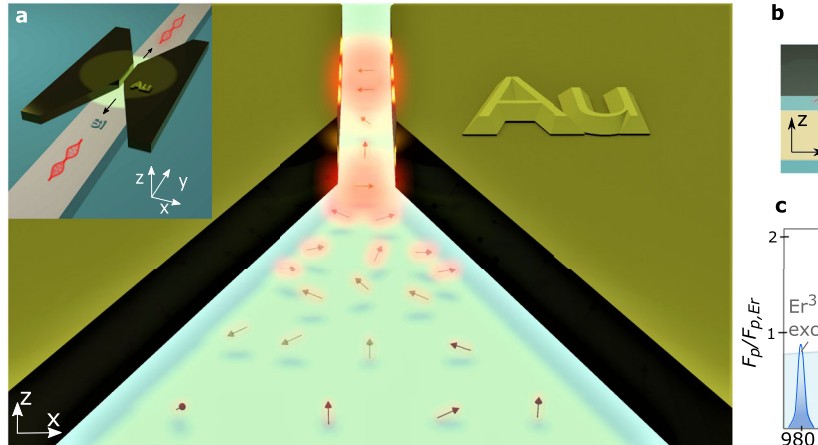

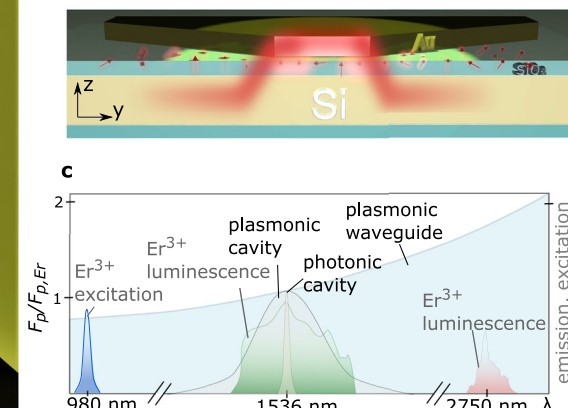

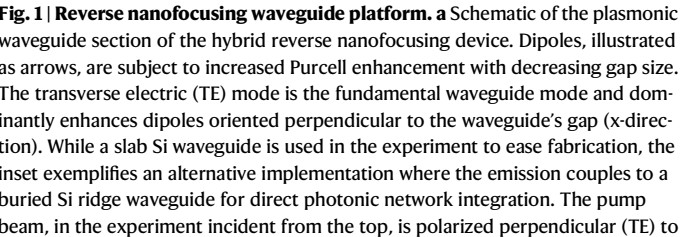

**Fig. 1 | Reverse nanofocusing waveguide platform. a** Schematic of the plasmonic waveguide section of the hybrid reverse nanofocusing device. Dipoles, illustrated as arrows, are subject to increased Purcell enhancement with decreasing gap size. The transverse electric (TE) mode is the fundamental waveguide mode and dominantly enhances dipoles oriented perpendicular to the waveguide's gap (x-direction). While a slab Si waveguide is used in the experiment to ease fabrication, the inset exemplifies an alternative implementation where the emission couples to a buried Si ridge waveguide for direct photonic network integration. The pump beam, in the experiment incident from the top, is polarized perpendicular (TE) to

the waveguide. **b** Waveguide cross-section illustrating the layer stack as well as the hybrid coupling of the dipole emission into the gap (emission in red) and the subsequent out-coupling into the underlying Si waveguide. **c** Comparison of Purcell enhancements in a plasmonic waveguide, plasmonic cavity ($Q$~10) and photonic cavity ($Q$~10⁵) as a function of wavelength[72]. The waveguide Purcell factor was calculated based on $F_p = 3c\lambda^2/(4\pi n^2 v_g A_m)$ as described in the text and normalized by $F_p$ at 1536 nm, $F_{p,r}$. The excitation and luminescence spectra of $Er^{3+}$ are qualitatively shown for comparison.

enhancement at par with or larger than photonic[7,47] and other plasmonic[48] waveguides, respectively.

## Results

### Quantum emitter-doped reverse nanofocusing waveguide platform

Figure 1a illustrates an emitter-doped hybrid gap plasmon waveguide system[29,30,49]. It allows efficient light extraction from $Er^{3+}$-ions placed within a highly confined gap into a photonic mode by reverse nano-focusing. Nanofocusing methods have been used and studied in great detail[28] to efficiently concentrate light far below the diffraction limit, but here the method is used in reverse. The platform enables interfacing plasmonically enhanced $Er^{3+}$-ions on the nm-scale, with the μm-scale photonic regime where emission can be guided through a low-loss silicon waveguide as illustrated in the inset. A sideview of the structure is given in Fig. 1b. We note that this device is not designed to operate as an antenna and simulations show only a moderate two-fold field intensity enhancement under the normal incidence excitation used in the experiment (Supplementary Information Fig. S1 a2, a3). Importantly, unlike plasmonic antenna emission[3], light is transported in plane to enable the separation of excitation and emission coupling.

The platform consists of a plasmonic tapered metal waveguide placed on an industry standard silicon-on-insulator (SOI) substrate separated by a thin $SiO_2$ layer, doped with erbium (Supplementary Information Figs. S2 & S3). In the experiment, the system is top-illuminated by linear polarized light focused onto its gap region. Erbium ions with electric dipole moments aligned to the gap mode's field distribution and direction (here TE) experience enhanced luminescence; hence, emission into this mode will dominate over slower radiative and non-radiative pathways. Fluorescence quenching due to coupling to higher-order plasmonic modes[50], is expected to be minor[50–52] with a slow non-radiative decay[53]. Following luminescence into the plasmonic mode, light subsequently expands in the low-loss

taper region. Taper angle and oxide thickness have been engineered to satisfy the eikonal approximation[21,29] enabling gap-to-silicon-slab out-coupling efficiencies of >80% (Supplementary Information Fig. S5b). Fig. 1c compares plasmonic waveguide and cavity Purcell factors as a function of wavelength. The luminescence spectra of erbium transitions at 1536 nm and 2750 nm for a 980 nm excitation wavelength are qualitatively shown for comparison. While resonant structures, such as antennas, require tuning of the cavity resonance and emission wavelength of the emitter, the non-resonant plasmonic waveguide offers broadband emission enhancement, collection and guiding with the potential to address all erbium transitions simultaneously without tuning.

Figure 2a–g present the characterization of the platform, shown in its measurement configuration in Fig. 2e. Curved grating couplers[29], which are matched to the circular wavefront of light emitted from the narrow gap waveguide, are employed to direct light from the silicon waveguide into free-space for detection. Figure 2a shows luminescence spectra as a function of pump wavelength corresponding to the characteristic spectral shape of $Er^{3+}$-ions in $SiO_2$ for a pump wavelength of 980 nm[54]. Figure 2b shows a lateral scan of the excitation beam along a $w = 10$ nm wide gap waveguide, confirming that the maximum signal stems from the gap region. The scan's spatial width corresponds approximately to the pump spot diameter ($\varnothing \approx 2$ μm). Figure 2c shows the luminescence signal for different gap lengths but at constant gap width. The exponential decay, attributed to mode propagation loss, confirms out-coupling via the gap waveguide mode. By using an iris diaphragm in the image plane, we were able to selectively detect light either from the whole device or from a single output grating and the waveguide region alone (second grating blocked). Here, the signal from one grating is about half of the total emission, which indicates that luminescence dominantly couples directly into the waveguide and is mainly directed into free-space via the gratings. The strongest signal, observed for a 1 μm long waveguide, represents a trade-off between

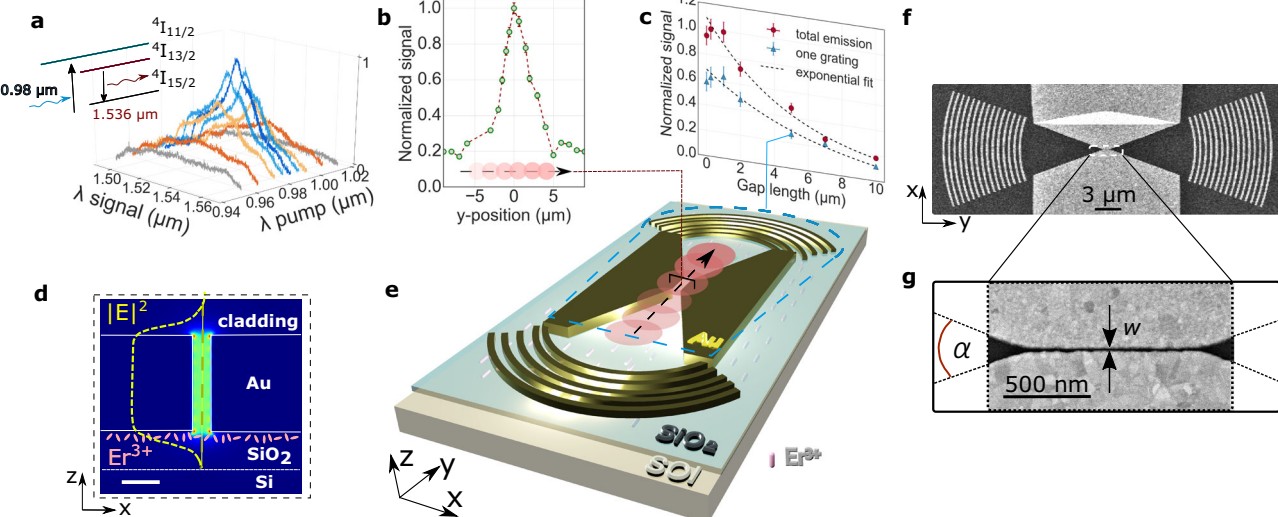

**Fig. 2 | Optical characterization of the erbium doped hybrid gap plasmonic waveguide.** The system is illustrated in its measurement configuration in **e**. **a** Measured luminescence spectra at different signal (λ signal) and pump wavelengths (λ pump) for a $w = 60$ nm wide gap waveguide. The inset illustrates the pump scheme and the main $Er^{3+}$- energy levels involved. **b** Luminescence signal dependence on the beam position, scanned along the waveguide as illustrated in (**e**), normalized to the maximum signal (measurement points). Dashed red line is a guidance for the eyes. Error bars are 3% of the measured value as extracted from the standard deviation of the $w = 10$ nm gap measurement. **c** Luminescence signal (measurement points) as a function of gap length at constant gap width, $w = 60$ nm, with total emission in red, and emission from only one out-coupling grating in blue.

The signals are normalized to the peak value of the total emission with exponential decay fits shown as dashed lines. Error bars are 8% of the measured value as extracted from the standard deviation of the $w = 60$ nm gap measurement. **d** Electric field intensity distribution $|E|^2$ of the waveguide mode in light green (50 nm Au, 25 nm $SiO_2$, 220 nm Si, PMMA cladding). Dashed yellow line overlay highlights the field intensity along the gap center and overlap with the underlying ions. Scale bar is 20 nm. **e**, $Er^{3+}$-doped hybrid waveguide platform consisting of a silicon-on-insulator (SOI) slab waveguide, a $SiO_2$ spacer layer and a gold waveguide with out-coupling gratings. **f**, **g** Scanning electron microscopy (SEM) top-views of a hybrid device with a waveguide gap length of $L = 1$ μm. The zoom-in discloses a $w = 10$ nm wide metal gap with opening angle $\alpha$.

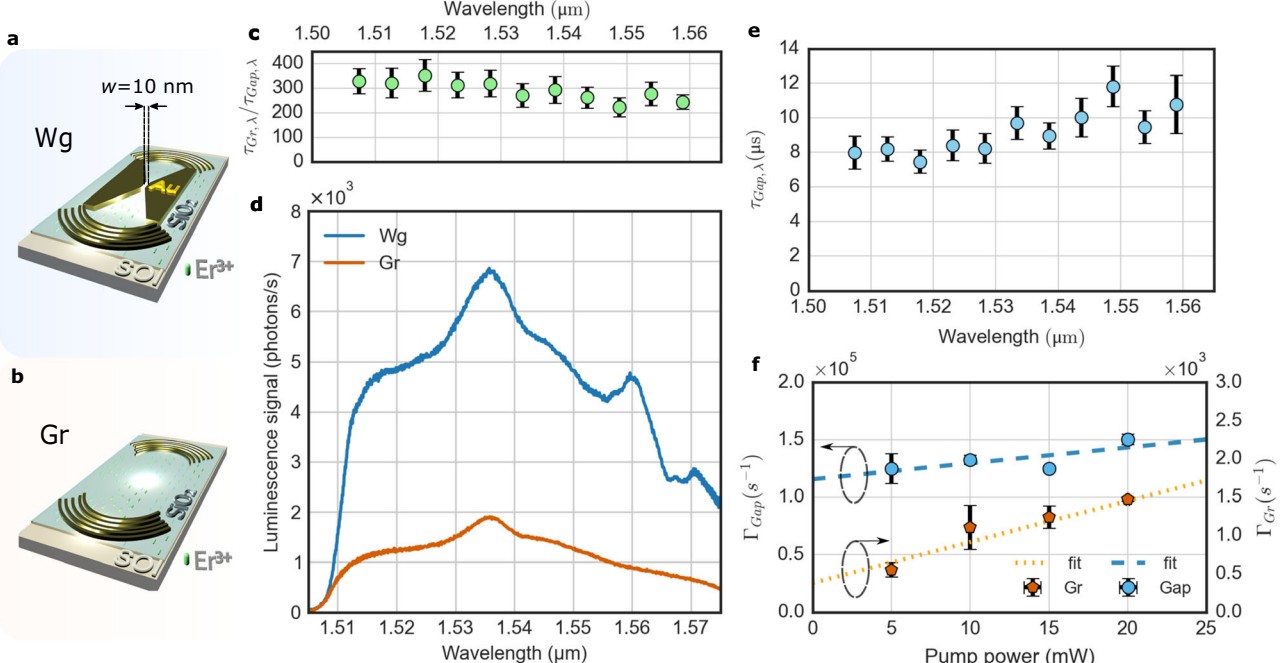

**Fig. 3 | Total emission rate and luminescence enhancement. a, b** Illustration of the hybrid waveguide (Wg) and the Gr-to-Gr configuration (Gr). Note that Wg device contains ions excited in the plasmonic gap and in the taper region, which are subject to different rate enhancements. **c,** Spectrally resolved lifetime ratio $\tau_{Gr,\lambda}/\tau_{Gap,\lambda}$ for the ions in the Gr device and the gap region of the Wg device with a $w = 10$ nm gap at zero pump power. Data points and error bars are the mean ± standard deviation. **d** Luminescence spectra measured for the devices shown in (**a, b**) at a pump power of 40 mW and an excitation wavelength of 980 nm in TE polarization (CW pump laser, $\varnothing \approx 2\,\mu m$) for 1 μm long waveguides (including taper and gap region of the Wg device). Note that the spectra have not been normalized

by the amount of illuminated ions. In fact, for the Wg device only a small fraction of the number of ions excited in the Gr device contribute to the signal. **e** Lifetime wavelength dependence for ions placed in the gap of the Wg ($w = 10$ nm) at zero pump power. Data points and error bars are the mean ± standard deviation. **f** Emission rates extracted from time-resolved luminescence spectroscopy averaged over the entire spectrum, comparing the Gr with the gap of the Wg device ($w = 10$ nm). The y-intercept yields the emission rates for the gap, $\Gamma_{Gap}$, and the grating device, $\Gamma_{Gr}$ at zero pump power as used in the text. The dotted and dashed lines are the linear fits for each case. Data points and error bars are the mean ± standard deviation.

propagation length and the number of excited Er³⁺-ions coupled to the plasmonic waveguide. The propagation length for the $w = 10$ nm waveguide is ≈2 μm as extracted from transmission measurements (Supplementary Information Fig. S5a). The plasmonic waveguide length was set to 1 μm for subsequent studies to limit collection losses. Figure 2d, showing the gap mode's electric field intensity distribution, reveals a good field overlap with the Er³⁺-ions implanted just below the Au/SiO₂ interface (yellow dashed profile overlay). Figure 2f, g show scanning electron microscopy top-views of the reverse nanofocusing device with a 10 nm wide and 1 μm long metal gap waveguide.

### Luminescence and emission rate enhancement

We first compare the erbium lifetime and luminescence brightness in the most confined plasmon waveguide with a control sample, shown in Fig. 3. Figure 3a, shows the hybrid metal-dielectric waveguide structure (Wg) of gap width, $w = 10$ nm, while Fig. 3b shows a control sample with only out-coupling gratings (Gr). Here, the control device is a reference for Er³⁺-luminescence that couples to the silicon slab and out of the gratings. A direct comparison of the spectrally resolved lifetimes for the gap section of the waveguide, $\tau_{Gap,\lambda}$, and the grating, $\tau_{Gr,\lambda}$, device, shown in Fig. 3c, yields a mean total emission rate enhancement factor of $\tau_{Gr,\lambda}/\tau_{Gap,\lambda} = \Gamma_{Gap,\lambda}/\Gamma_{Gr,\lambda} = 297 \pm 36$. The enhancement is relatively uniform across the measured spectrum. The total emission rates of ions in the gap, $\Gamma_{Gap} = \gamma_R + \gamma_{NR}$, and in the grating control, $\Gamma_{Gr} = \gamma_{0,R} + \gamma_{0,NR}$, have radiative ($\gamma_R$) and non-radiative ($\gamma_{NR}$) contributions, while the Purcell effect, $F_p$, only acts on radiative transitions, i.e. $\Gamma_{Gap} = F_p\gamma_{0,R} + F_{NR}\gamma_{0,NR}$. Here $F_{NR}$ accounts for the change of non-radiative emission due to the metal. It is important to verify that the device does not only exhibit faster recombination, due to $F_{NR}$ and/or

$F_p$, but also brighter emission due to $F_p$. This is demonstrated in Fig. 3d, showing that lifetime reduction is also accompanied by a strong luminescence enhancement compared to the control. Note that the spectra here are not normalized to the number of contributing ions. The gold regions in the waveguide device mask the pump light to excite <12% of the ions that contribute to the grating device measurement. We will see later that about 80% of the signal from the waveguide device originates from the gap, representing just 0.3% of excited ions in the grating device. This makes the gap ions of the waveguide device several orders of magnitude brighter than ions in the grating control device. The reduced erbium lifetime combined with increased brightness suggest that the radiative Purcell enhancement dominates.

Figure 3e spectrally resolves the ion lifetime in the gap of the waveguide device to show a range, $\tau_{Gap,\lambda} = 8\,\mu s{-}10\,\mu s$. Meanwhile, lifetimes of the grating control sample were ≈3 ms, while the natural lifetime of Er³⁺-ions implanted into a SiO₂ host matrix is typically tens of ms[54,55]. Here, the presence of a Si slab[55] and high doping concentrations[56] can explain such a lifetime reduction. Nonetheless, our measurements demonstrate an erbium emission rate approaching three orders of magnitude faster than the natural lifetime of erbium in SiO₂. $\tau_{Gap,\lambda}$ was determined by extrapolating the lifetime measured at each wavelength and for various powers to zero power excitation, excluding power-dependent effects. Additionally, Fig. 3f provides the emission rates, $\Gamma_{Gap} = \tau_{Gap}^{-1}$, for the $w = 10$ nm waveguide and, $\Gamma_{Gr} = \tau_{Gr}^{-1}$, for the grating device. These were averaged across the spectrum for each excitation power. The spectrum averaged extracted lifetimes are $\tau_{Gap} = 8.6\,\mu s \pm 0.8\,\mu s$ for the ions in the gap of the waveguide device and $\tau_{Gr} = 2.6$ ms ± 0.7 ms for the grating device, which results in an

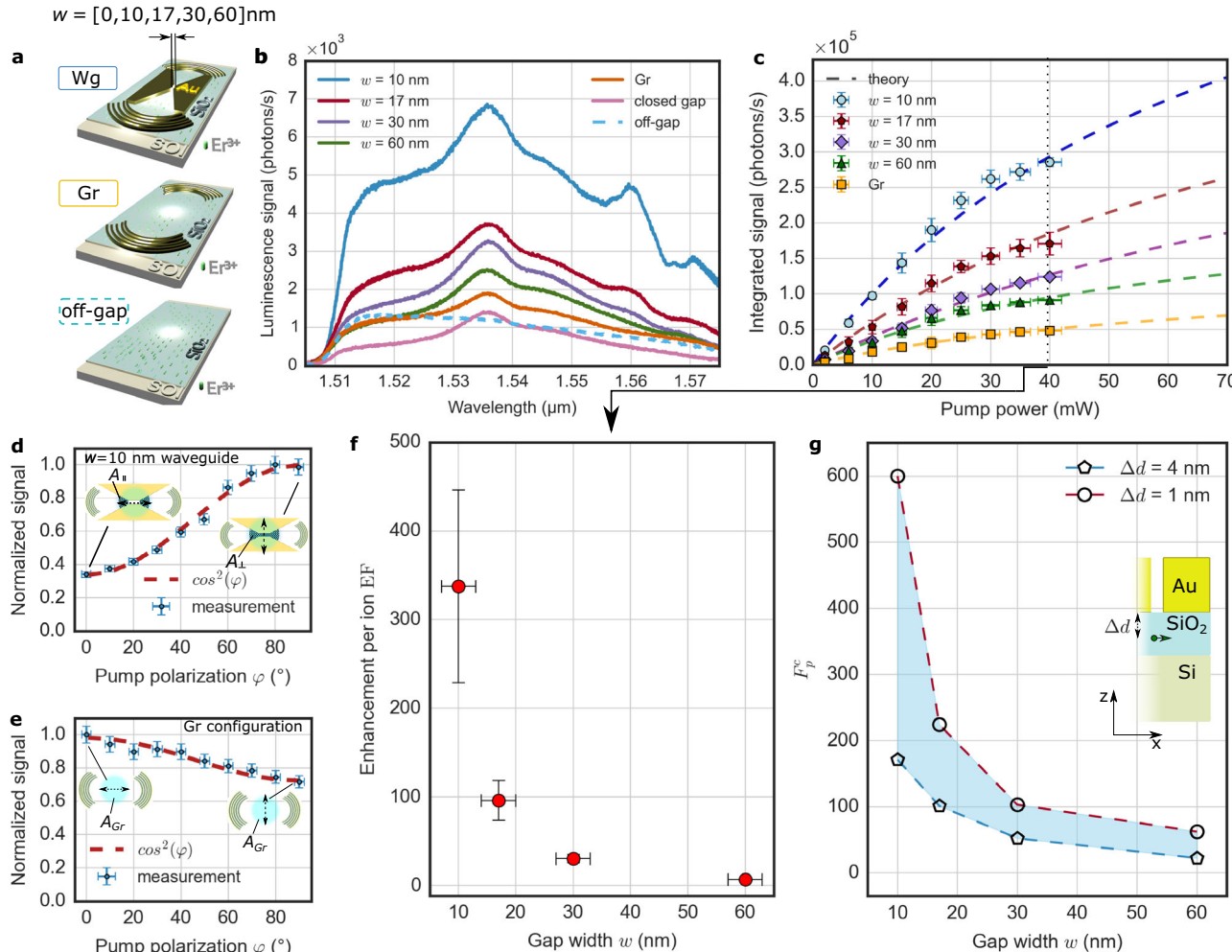

**Fig. 4 | Luminescence spectra, power dependence and luminescence enhancement factor. a** Illustration of the compared devices: hybrid gap plasmon waveguide system (Wg, described in Fig. 1h), only out-coupling gratings (Gr) and substrate with implanted ions (off-gap) Illumination. Beam spot in white. **b** Luminescence spectra measured for the devices shown in (**a**) with different Wg gap widths $w = [0, 10, 17, 30, 60]$ nm ($w = 0$ nm for closed gap) at a pump power of 40 mW and an excitation wavelength of 980 nm from the top in TE polarization (CW pump laser, $\varnothing \approx 2\,\mu m$) for 1 μm long waveguides. Note that the common Si-background has not been subtracted at this point. It contributes about half of the "off-gap" signal (Fig. S6). **c** Laser pump power dependence of the integrated spectra from (b) with corresponding theoretical fits (Supplementary Eq. (S6.5) and Eq. (S6.8)) based on the three-level rate equation model (Supplementary Information S5 & S6). The extracted values for the saturation signal and power are listed in Supplementary Information Table S1. Data points and error bars are the mean ± standard deviation. Note that the samples were not subject to intensities that could

cause any changes in morphology (i.e., damage). **d, e** Dependence of emission on pump beam polarization. Central illumination with changing polarization angle $\varphi$ at a wavelength of 980 nm for (**d**) the Wg device with a $w = 10$ nm gap waveguide and (**e**) the Gr control device. The double arrows indicate the polarization of the pump beam while $A\|$ and $A\perp$ in (**d**) refer to the illuminated area in the different polarization configurations (dark shaded area) and $A_{Gr}$ in (**b**) is the focal spot area. Error bars are estimated to 5% of the measured value for the waveguide and the Gr configuration based on the standard deviation determined in (**c**) for the same power. **f** Luminescence enhancement factor per ion EF at 40 mW for various gap widths extracted from (**g**) and Eq. 1. Error bars are calculated by error propagation of Eq. 1. The Gr configuration is used as reference. **g** Simulated average Purcell factor $F_p^c$ obtained from averaging over different positions across the gap for a dipole emitter placed at distance $\Delta d = [1\,nm, 4\,nm]$ below the gap waveguide (see inset, green arrow). Here, only dipoles aligned perpendicular to the gap are considered.

emission rate enhancement of $\Gamma_{Gap}/\Gamma_{Gr} = \tau_{Gr}/\tau_{Gap} = 306 \pm 83$. The mean of the spectrally resolved emission rate agrees well with value extracted from the spectrally averaging. We now continue to quantify the luminescence brightness enhancement per ion to determine an estimate for the Purcell factor, $F_p$.

Figure 4a shows three device configurations studied to quantify the luminescence enhancement in the nanofocusing waveguide. As in Fig. 3, we consider plasmonic gap waveguide structures (Wg), now with varying gap width, $w$, and the control sample with gratings only (Gr). We also consider regions of the sample without metallic structures (off-gap), as a second control measurement for the proportion of Er$^{3+}$-luminescence radiated directly to free space. Note that the $w = 0$ nm case represents another waveguide control device

(closed gap), to assess the Er$^{3+}$-luminescence from the triangular taper regions on either side of the 1 μm long gap waveguide.

Figure 4b shows the luminescence spectra for waveguide devices of varying gap width $w$, and the three control devices: Gr, off-gap and closed gap. Importantly, we observe a net emission enhancement for all non-zero gap widths compared to the control devices. The emission enhancement occurs despite the metallized regions masking the number of Er$^{3+}$-ions that can be excited and also despite plasmonic propagation or coupling losses. Moreover, the luminescence continues to improve with reducing gap width, even with the reduction in the number of Er$^{3+}$-ions contributing to the emission.

Figure 4c shows the pump power dependence of the integrated spectra for the various gaps of the waveguide ($S_{Wg}(w)$) and the grating

$(S_{Gr})$ configuration. Here, a Si-substrate background signal was subtracted as explained in Supplementary Information S4. The integrated signal increases with decreasing waveguide width for all powers. The observed luminescence enhancement may arise due to both improved excitation as well as the Purcell effect. To disentangle the two processes, we consider the saturation behaviour of the $Er^{3+}$-ions. At saturation, the rate of $Er^{3+}$-ion luminescence is independent of the excitation power and is thus determined only by the emission rate itself. The emission from $Er^{3+}$-ions is commonly described by three-level rate equations[37,57,58], which forms the basis for a saturation model (Supplementary Information S6). The model (dashed lines in Fig. 4c) shows a good agreement with the experimental data (Supplementary Information Tab. S1).

To determine the contribution of ions in the gap, $S_{Gap}(w)$, to the total waveguide signal, $S_{Wg}(w)$, we use the control measurement of the closed gap waveguide ($w = 0$) to assess the signal from the taper region. We further corroborate this estimate using the polarization dependence of luminescence; only pump light polarized perpendicular to the gap may excite ions below the gap (Supplementary Information S1 & S7). This allows to switch the plasmonically enhanced ions on ($\varphi = 90°$) and off ($\varphi = 0°$). Figure 4d, e plot the integrated $Er^{3+}$-signal for the 10 nm gap waveguide and the grating device as a function of pump polarization angle, $\varphi$, respectively. For example, both methods, outlined in Supplementary Information S7, indicate that ≥78% of the total waveguide signal originates from ions in the gap alone ($S_{Gap}(w = 10 \text{ nm}) \geq 0.78 \, S_{Wg}(w = 10 \text{ nm})$).

It is now possible to directly extract the Purcell factor, $F_p$, of ions in the gap from the saturation curve fits in Fig. 4c; however, in order to provide an estimate for $F_p$ that does not rely on any assumptions made in the model, we use the measured data points $S_{Gap}^{40mW}(w)$ and $S_{Gr}^{40mW}$ at a pump power of 40 mW. Since the ratio of the integrated luminescence signals, $S_{Gap}(w)/S_{Gr}$, monotonically increases with pump power, the ratio at 40 mW provides a lower bound estimate of $F_p$ (Supplementary Information S5, Eq. S5.29). This lower estimate, here called luminescence enhancement factor, EF(w), is given in Fig. 4f as a function of gap width[59]:

$$\text{EF}(w) = \frac{S_{Gap}^{40mW}(w)}{S_{Gr}^{40mW}} \frac{A_{Gr}}{L \cdot w} \frac{\varepsilon_{c,Gr}}{\varepsilon_{c,Gap}(w)}, \quad (1)$$

where, $L$ is the waveguide length, $w$ is the gap width, $A_{Gr}$ is the area of the illumination beam as illustrated in Fig. 4e. The signal collection efficiencies from the excited regions on the sample are $\varepsilon_{c,Gap}(w)$ for the waveguide device and $\varepsilon_{c,Gr}$ for the grating device, estimated from simulations (Supplementary Information S8).

We find a lower bound estimate for the Purcell enhancement factor of $F_p > \text{EF} \approx 338 \pm 109$ for a $w = 10$ nm wide gap. This value is commensurate with the lifetime reduction factor for this waveguide and indicates that the radiative enhancements of the plasmonic waveguide outweigh the non-radiative ones. The large Purcell factor is comparable to those achieved in high-Q cavities[47,60], while our approach in contrast is broadband and allows for fast photon collection. The corresponding values used to arrive at the estimate from Fig. 4f are listed in Supplementary Table S2.

Figure 4g shows the computed Purcell factor, $F_p^c$ (Supplementary Information S6), for varying gap widths and implantation depths, $\Delta d$. The simulated ion depths (inset Fig. 4g) cover a distance range of $\Delta d = 1$ nm to 4 nm below the $SiO_2$ surface. Although the implantation depth distribution reaches the maximum at 6 nm (Supplementary Information Fig. S3), a peak ion implantation depth of about 2 nm is estimated when taking annealing diffusion towards the $SiO_2$ surface into account[61] (Supplementary Information S2). Simulations show that Purcell factors up to 600 for a 10 nm gap waveguide are feasible with the proposed platform.

## Addressing electric dipole transitions

We now turn the reader's attention to the shape of the luminescence spectrum shown in Fig. 5a for a $w = 10$ nm waveguide device. In addition to the 1.536 μm transition, peaks at 1.557 μm, 1.560 μm, 1.567 μm and 1.571 μm become visible. The top right inset in Fig. 5a shows the normalized spectra for $w = 10$ nm, $w = 17$ nm and a grating control device, confirming a change in spectral shape with gap width. In general, local electrical fields in a $SiO_2$ host matrix at the $Er^{3+}$-sites lead to Stark splitting of the $^4I_{13/2}$ and $^4I_{15/2}$ energy levels[27,62,63]. At room temperature, these energy level transitions are typically not resolved and merely broaden the spectrum[54]. Meanwhile, the electric dipole (ED) and magnetic dipole (MD) transitions of $Er^{3+}$-ions overlap spectrally and are of similar magnitude[64,65]. Here, we resolve the electric dipole transitions due to the strengthened coupling of the $Er^{3+}$-ion electric dipole to the plasmonic gap mode, while magnetic dipole transitions are not enhanced. The smaller the gap width, the smaller the effective mode area and the stronger the confinement of the TE gap mode. This results in a large electric local density of states to which light, emitted from ED transitions, couples best[65]. Note, that

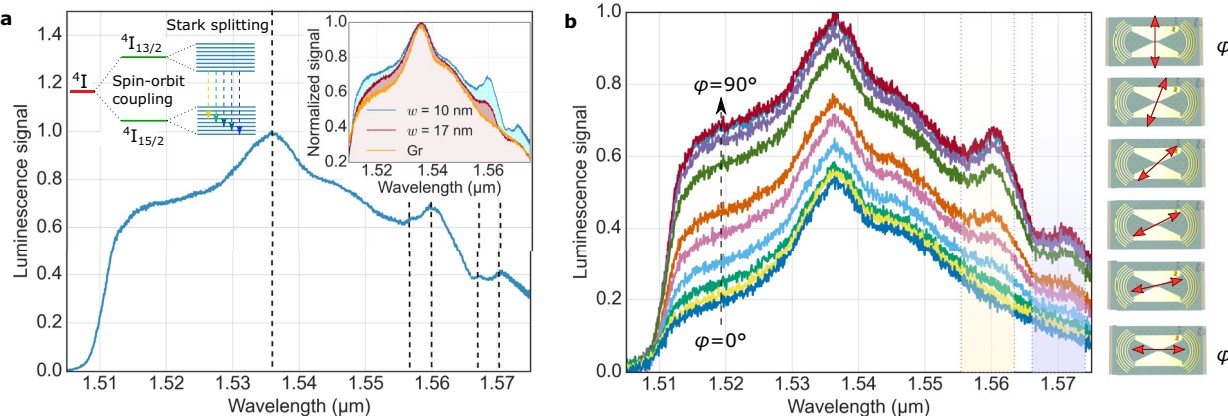

**Fig. 5 | Stark splitting and polarization dependence. a** Measured normalized luminescence spectra for a $w = 10$ nm Wg at 40 mW CW pump power. Dotted lines mark the energy transitions observed in the experiment. The top left inset schematically illustrates the Stark splitting and the resulting transitions between the $^4I_{13/2}$ and $^4I_{15/2}$ energy levels. The top right inset compares the normalized spectra of a $w = 10$ nm (blue) with a $w = 17$ nm (red) waveguide and the signal measured in the Gr (orange) device. **b** Polarization dependent luminescence spectrum for the same settings as shown in (**a**). The schematics on the right illustrate the polarization angle $\varphi$ of the pump beam with respect to the gap waveguide orientation. The luminescence increases monotonously with increasing angle. Yellow and purple transparent overlays highlight the emerging stark transitions.

the transition peaks observed in Fig. 5a (dashed lines) can be identified at cryogenic temperatures[66]. Simulations confirm that MD emission enhancement by the plasmonic gap is negligible (Supplementary Information S9). The exact energy level positions depend on environmental parameters which could be studied in the future. Figure 5b shows pump beam polarization dependent measurements for the $w = 10$ nm waveguide device. Transitioning from perpendicular to parallel polarization relative to the gap turns the additional peaks off, as the incident beam no longer excited erbium ions in the gap. The ability to selectively access and control multiple ED transitions over a broad frequency range is of interest for a wide range of materials and technologies[67].

## Discussion

In summary, we have presented an approach to locally enhance erbium ion emission on the nanoscale as well as extract and guide emitted light to a low loss photonic waveguide. Efficient plasmonic-to-photonic mode conversion was achieved by means of a hybrid plasmonic-photonic reverse nanofocusing technique. The reverse nanofocusing hybrid plasmonic waveguides exploit the Purcell effect to simultaneously increase the total emission rate of $Er^{3+}$-ions by a factor of $306 \pm 83$ and the radiative rate by a factor of $338 \pm 109$. The non-resonant nature of the plasmonic waveguide allows for Purcell enhancement across the erbium emission spectrum, which strongly relaxes the need for mutual emitter-cavity tuning commonly required in high-Q cavities.

Our platform enabled the enhancement of the Stark split electric dipole transitions of the $Er^{3+}$-luminescence band at room temperature, showcasing the ability to access and enhance multiple electric dipole transitions by a single plasmonic waveguide mode despite phonon broadening. The hybrid plasmonic waveguide platform allows for simultaneous luminescence enhancement and guiding, while relying on facile and silicon photonic compatible fabrication. This is important for the development of quantum technologies using single emitters, such as single photon sources[7,68], and ensembles of ions, such as quantum memories[33]. Our approach does not limit the choice of emitter materials, opening up a range of possible frequency regimes and applications for luminescence enhancement such as on-chip communication and sensing. The possibility to directly contact and control the waveguide electrically provides a simple method to manipulate transition states, such as via the Stark effect, bearing possibilities for opto-electrical signal modulation of quantum emitters.

## Methods

### Sample fabrication

An outline of the individual fabrication steps of the reverse nanofocusing hybrid gap plasmonic waveguides with implanted $Er^{3+}$-ions is illustrated in Fig. S2 while top-view SEM images of the resulting structures are depicted in Fig. S2b–e. First, a 25 nm $SiO_2$ layer was sputtered onto commercially available SOI wafers. After that, Erbium-ions were implanted into the $SiO_2$ interlayer at room temperature using ion implantation with an acceleration voltage of 10 keV, a sample tilting angle of 45° and an ion fluence of $\rho = 1 \times 10^{15}$ cm$^{-2}$. This resulted in a mean ion implantation depth of $(6.0 \pm 2.5)$ nm, as simulated with the Monte-Carlo software package TRIM[69–71]. This is followed by an activation step in which the samples were annealed under controlled atmosphere for one hour at 900 °C. Here, directional ion migration during the annealing process towards the $SiO_2$ interlayer surface can take place (estimated as $\langle y \rangle \approx 3.8$ nm), shifting the ion implantation density peak closer to the surface (Supplementary Information 2) as reported in the literature[61]. After annealing, hybrid gap plasmonic waveguides (Wg) were fabricated on top of the $Er^{3+}$-doped $SiO_2$ interlayer. To produce narrow metal-insulator-metal (MIM) gaps, a two-step electron beam lithography (EBL) process with a large range of varying target gap widths has been used resulting in gaps as small as 10 nm.

After each EBL step, a 50 nm Au layer was deposited via evaporation at ambient temperature and at a pressure $<5 \times 10^{-7}$) Torr. Finally, the samples were covered by a thick cladding layer of poly(methyl methacrylate) (PMMA).

### Optical measurements

The optical characterisation was performed using the setup illustrated in Supplementary Information S10. A CW laser diode at a wavelength of 980 nm was used to pump the erbium ions in the metal gap waveguides. The angle of linear polarization of the incident light was controlled using a $\lambda/2$ - waveplate and a polarizer. Here mostly perpendicular to the gap polarized light was used (TE polarization) to excite the ions below the gap most effectively. The pump beam was focused onto the centre of the plasmonic gap by an infinity-corrected achromatic near-IR objective (NA = 0.4, 20$x$, WD = 2 cm). The exact waveguide position with respect to the beam was controlled via a closed loop piezo stage with an IR camera used to image the sample and spot positions. The measurement was performed in reflection and the erbium ion signal was collected via the same objective after coupling to free space via curved grating couplers which were optimized for a centre wavelength of 1.536 μm and TE polarized emission. The collected and collimated erbium signal was analysed using a nitrogen cooled (−120 °C) InGaAs camera spectrometer (Princeton Instruments Spec 10 system with a SP2300). The spectrometer grating used had a groove density of 600 gr/mm and a blaze wavelength of 1.6 μm with an efficiency of ≈85% at 1536 nm. To select specific regions of the sample, the collected light was focused onto an image plane in which an adjustable iris was positioned. This allowed the signal from only one coupling grating to be collected. However, the main measurements were performed without the pinhole in order to collect the maximum amount of light from the sample via both gratings. Polarization filters were employed to control polarization and power, while suitable short and long pass filters, as shown in Fig. S13, were used to ensure that no potential additional pump signal reached the detector. All measurements were performed in the dark at a maximum integration time of 120 s.

A second, independent estimate of the luminescence enhancement in the waveguide system was provided based on time resolved spectroscopy. Here, a modulated pump and detection technique as described in further detail in the Supplementary Information 11, using a 980 nm CW pump laser and a higher NA objective to collect the gap signal alone, was employed. The incoming pump as well as the outgoing signal beam are modulated by two phase-locked, out of phase (π-shift) mechanical choppers in order to single out the signal decay in time as illustrated in Supplementary Information Fig. S14. Advantages of this technique are that it allows the elimination of power dependent effects common to ultrafast pulsed techniques and enables measurements of spectrally-resolved emission rate enhancement over a broad wavelength range.

### Simulations

Finite-difference-time-domain (3D FDTD) simulations were used to calculate the electric fields within the nanogap waveguide (c.f. Fig. 1), optimize in- and out-coupling efficiency, determine the polarization dependence of incident light as well as the numerical Purcell enhancement estimate. For the latter, the $Er^{3+}$-ion was modelled as an electrical dipole emitter with a perpendicular orientation of the dipole moment to the gap (TE), at a wavelength of 980 nm. The emitted power was determined by a transmission box and a homogeneous mesh of sub-nm mesh cell size. The estimate for the Purcell enhancement resulted from dividing the total emission of a dipole placed below the gap waveguide for various gap widths by the transmission resulting from an identical simulation setting in a reference simulation without Au waveguide structures (Gr device). The magnetic dipole in

Fig. S12 was simulated in the same manner. In order to estimate the vertical radiation in Wg and Gr configuration towards the objective as well as the power coupled into the Si slab from a horizontal dipole, the transmitted power in each direction was calculated and divided by the total emitted power in all directions from an electric dipole aligned parallel to the pump beam polarization, in analogy to the experiment.

## Data availability

The data that support the findings of this study have been deposited in the Research Data Repository of Imperial College London and are available online at 10.14469/hpc/12430. Any additional materials and data are available from the corresponding author upon request.

## Code availability

The code used to perform the FDTD simulations supporting the findings of this study is available from the corresponding author upon request.

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

## Acknowledgements

This work was supported by the EPSRC Reactive Plasmonics Programme (EP/M013812/1), the EPSRC Catalysis Plasmonics Programme (EP/W017075/1) and the Leverhulme Trust (RPG-2016-064) (R.F.O). The authors thank Anita Chandran and Robert T. Murray from Imperial College London for providing valuable laser equipment and support. We like to thank the German National Academy of Sciences Leopoldina for their support via the Leopoldina Postdoctoral Fellowship (LPDS2020-12) (N.A.G), the Deutsche Forschungsgemeinschaft (DFG) for financial support within the frame of the collaborative research center CRC 1375 "Nonlinear optics down to atomic scales (NOA)", project C5 (C.R). We additionally acknowledge the Lee-Lucas Chair in Physics (S.A.M). This project has received funding from the European Union's Horizon 2020 research and innovation programme under a Marie Skłodowska-Curie Fellowship (grant agreement no. 844591) (M.F.).

## Author contributions

N.A.G., R.F.O. and C.R. conceived and designed the experiments. N.A.G. fabricated and characterized the samples. N.A.G. performed the experiments and analyzed the data. M.F. designed, performed and analyzed the time-resolved luminescence spectroscopy. N.A.G. performed the simulations. M.Z. performed the ion-implantation and implantation depth calculations. M.P.N. and P.D. assisted with preliminary characterizations of samples. N.A.G. wrote the manuscript with input from R.F.O. as well as M.F., M.Z., M.P.N., P.D., R.R., A.S.C., S.A.M., and C.R., R.F.O. and S.A.M supervised the project.

## Competing interests

The authors declare no competing interest.
