## [Peer Review File · Nature Communications]

Emission enhancement of erbium in a reverse nanofocusing waveguideREVIEWER COMMENTS

Reviewer #1 (Remarks to the Author):

The manuscript by N.A. Gsken et al reports a reverse nanofocusing waveguide for enhancing the emission rates of erbium ions. This waveguide is shown to increase the broadband emission rate of erbium ions by a factor of 306 and increase the radiative rate by a factor of 338 at maximum. It allows the observation of stark splitting at room temperature.

Overall, the authors have done adequate experiments to show the performance of their device. The design of the experiments and the following analysis are rigorous and convincing. Since the non-resonant structure eliminates the requirement of precise matching the wavelength of the emitter and the cavity, it will for sure facilitate high-yield, large-scale production of quantum emitters. Therefore, I would be happy to recommend publication of this work.

A few points should be addressed in the revision.

1. Could the authors visualize their device performance (Purcell factor, bandwidth, etc) among other reported results? So far, I have only found such comparison in line 246 of the main text.
2. Figure 1c plots waveguide vs. plasmonic cavity Purcell factor as a function of wavelength. As the author already mentioned, the Purcell factor of cavity is determined by the Q and mode volume. Are the parameters used to plot Figure 1c representative? A fair comparison between resonant and waveguide structures, not limited to plasmonics, should be presented.
3. In Figure S5 b, the two simulated coupling efficiencies for 60nm gap width are obviously different. Please clarify.

Reviewer #2 (Remarks to the Author):

The manuscript by Gusken et al. demonstrates coupling of erbium ions to an engineered plasmonic nanowaveguide, showing Purcell radiative rate enhancements of ~ 300 for ensembles measured at room temperature. The plasmonic geometry includes both a gap nanowaveguide and a tapered nanofocusing region that allows efficient coupling of the former with a silicon slab waveguide located underneath the metallic region. I find the geometry promising, considering it can in principle provide relatively large Purcell enhancements alongside efficient transition into low-loss photonic waveguides.

The particular geometry demonstrated here is novel regarding quantum emitter coupling (it has been originally developed by the authors and reported in prior publications). However, experiments by other groups (e.g. Dibos et al. PRL 120 243601 (2018)) have already demonstrated larger or comparable Purcell enhancement factors as observed here, using photonic structures. The main advantage that is claimed in the paper - broadband operation - is compelling, however somewhat limited in appeal. For instance, slow light photonic crystals (as discussed in Ref. [1]), and dielectric bowties such as in ACS Photonics 9 1647 2016) offer similar advantages, and have shortcomings that are arguably at the same level as those of the present geometry. I think more context could have been provided to clarify the appeal of the work, particularly, but not limited to, regarding applications in quantum networks (for instance, which applications or measurements more specifically could such performance enable, in view of what has been demonstrated? Also, are there other unique characteristics, besides broadband operation, that could enabling?).

The methods are reasonable in principle, however further clarification is needed as indicated in my comments below. The text needs some work as well for clarity. For these reasons I feel I am unable to recommend publication in Nature Communications. It might be acceptable if my concerns above about the general appeal are properly addressed, as well as my comments below.

1 - Fig. 1a is somewhat misleading because the fabricated devices do not feature an etched photonic waveguide as depicted, but rather a silicon slab. I think the fact that a different geometry was tested should be better clarified in the text, including in the Figure 1 caption. I also think that the reason for building the geometry in Fig.1a should be expanded in the introduction, as well as the reasoning for utilizing the slab, rather than the waveguide geometry, in experiments. Lastly, what is the simulated dipole coupling efficiency into the ridge silicon waveguide? Is it as good as the claimed coupling into the silicon slab mode?

2 - On line 118, it is claimed that the emission into the plasmonic gap mode dominates over nonradiative pathways. I think the authors should expand this discussion, with some quantification. What nonradiative pathways can be reasonably expected, and what are their time-scales? The transitions that are targeted are fairly slow (\sim ns), and the enhancements that are expected only bring them to the \sim μ s scale. Are these time-scales sufficient?

3 - More experimental details should be given in Supplementary Section 3, regarding propagation length and coupling efficiency estimates. The text is somewhat confusing and the explanations not sufficiently clear. From the main text, I understood that propagation losses were obtained by measuring waveguide-outcoupled PL intensities for waveguides of varying lengths, for the same pump level. However, the SI text suggests that a passive cut-back transmission measurement was performed for such estimation, where a 1550 laser beam was launched into one grating and the transmitted beam power was measured.. Is that the case? The authors should make this more clear in the SI text.

Importantly, though, the coupling efficiencies $\eta_{\text{[inc,outc],gap}}$ in this case only correspond to the coupling efficiency between gap waveguide mode and the silicon slab. It does not correspond to the coupling efficiency of the Er emitters into the slab, because, strictly, the emitter may couple to radiative (and also possibly bound) modes, in addition to the gap mode.

The authors should provide simulation results quantifying the expected dipolar coupling efficiency to the plasmonic gap mode, and/or to the slab waveguide after the taper. This is mostly to rule out, theoretically, the possibility that coupling to unbound modes is significant and possibly comparable to the coupling rate to the (bound) waveguide mode. This coupling efficiency, the so-called modal β -factor, is very important, because it indicates how ultimately efficient the light-matter interface can be. Importantly, all measurements done to characterize such coupling only takes into account upwards emitted light, disregarding dipole emission that exits through the substrate. The waveguide-coupled light, then, is not the total that is emitted (though might be a good portion of it).

4 - Have the authors tried to perform transmission measurements with the Er³⁺-doped gap waveguides? It would be very interesting to see if a significant enhancement of the single-ion extinction cross-section due to the gap mode can be observed. The extinction has a direct dependence on the β -factor (see e.g. Turchmann et al., Nanophotonics 8 1641 2019).

5 - What accounts for the shorter lifetime observed from the grating device (\sim 3 ns) in contrast with the lifetime of bare Er³⁺ ions implanted in SiO₂?

6 - Can the authors further clarify the difference between $\tau_{\text{Gr},\lambda}$ and τ_{Gr} , and similar for the gap devices? Two enhancement values, $\tau_{\text{Ggap},\lambda}/\tau_{\text{Gr},\lambda}$, and $\tau_{\text{Ggap}}/\tau_{\text{Gr}}$ are given in the text (lines 162 and 188 respectively), and I don't understand exactly why. As far as I can tell, they are obtained for the same device, under the same conditions, and are enhancement averages over all probed wavelengths.

7 - What accounts for the linear increase of the emission rates with the pump power in Fig. 2f ? For what pump powers were $\tau_{Ggap,\lambda}$ / $\tau_{Gr,\lambda}$, and/or τ_{Ggap} / τ_{Gr} obtained?

8 - In the caption of Fig. 3 there is a comment that pump powers were limited to avoid changes to device morphology. Can the authors expand on what takes place and why?

9 - The difference in gap waveguide in/out coupling (quantified by $\eta_{inc,gap}$ and $\eta_{out,gap}$) is somewhat curious, I think the authors should expand on this somewhat. Since the structure is completely symmetric, I would expect in- and out-coupling to be reciprocal, unless there are additional input or output modes that are somehow involved. Can the authors provide more details about how the simulations to calculate this were performed?

10 - In Fig. 1a, why does the off-gap spectrum not display the same characteristic peak as the other curves? Is the spectrum just due to the SOI background emission, as in Fig. S6?

11 - In Fig. S1, normal excitation pump leads to a standing-wave in the gap. Why is that? Does this mean that the excited ions are only those located within the hotspots? Is this taken into account in the gap ion contribution ($S_{Gap}(w)$) estimation?

12 - On Fig. 3c, the luminescence intensity curves show signs of saturation at increasing powers, however a clear saturation level is not apparent. Because of this, it is difficult to know at what level the ion populations in the waveguide and reference (no-waveguide) samples are being pumped in their respective saturation curves. I think the authors should estimate, based on the Fig. 3c data, at what level of saturation the populations in the waveguide and reference samples are, for the 40 mW power level considered for extracting EF.

13 - The authors should expand the discussion of why the presumed Stark-split peaks are visible in the present experiments due to enhancement of ED transitions. In particular, the role of MD transitions in masking ED transitions in non-enhanced samples would be better clarified - are the two types of transitions expected to overlap spectrally in the SiO₂ host, and have the feature comparable transition probabilities?

Reviewer #3 (Remarks to the Author):

In this manuscript, the authors described emission enhancement by using a plasmonic nanowaveguide based on Purcell effect. The authors selected Er-ion doped platform at telecommunication wavelengths, which is of broad interest to the photonic community. The enhancement effect is convincingly measured and presented, and the authors built a theory to explain some of the behaviors. Overall, this is a paper of high quality, and the topic is interesting to the broad audience of photonics. I recommend it for publication in Nature Communications.

I only have two minor comments:

(1) In figure 2, it is difficult to understand why the large enhancement effect on lifetime in panel (c) only translates to a few times enhancement of luminescence in panel (d). Could the authors provide a simple, intuitive explanation for this?

(2) In figure 2 (a), the structure is labeled as "Wg," but in panel (c) and (e), the authors use "gap" to label results for the waveguide. This is a little bit confusing. I feel it's better to use "wg" everywhere throughout the manuscript.

Reviewer #1 writes: *The manuscript by N.A. Güsken et al reports a reverse nanofocusing waveguide for enhancing the emission rates of erbium ions. This waveguide is shown to increase the broadband emission rate of erbium ions by a factor of 306 and increase the radiative rate by a factor of 338 at maximum. It allows the observation of stark splitting at room temperature.*

Overall, the authors have done adequate experiments to show the performance of their device. The design of the experiments and the following analysis are rigorous and convincing. Since the non-resonant structure eliminates the requirement of precise matching the wavelength of the emitter and the cavity, it will for sure facilitate high-yield, large-scale production of quantum emitters. Therefore, I would be happy to recommend publication of this work.

Our Response:

We are pleased to read about the overall positive feedback. Further, we would like to thank the reviewer for the time and effort spent to read and revise our work. The comments are indeed helpful and will certainly improve the manuscript. We have addressed all comments in our replies (in green text) below, including modifications to the manuscript (in blue text):

Reviewer #1 writes: *A few points should be addressed in the revision.*

- *Could the authors visualize their device performance (Purcell factor, bandwidth, etc) among other reported results? So far, I have only found such comparison in line 246 of the main text.*

Our Response:

We have added a direct comparison to the introduction which puts our device into context with photonic and plasmonic resonators as well as waveguiding structures. Please see line 105 –116 of the main manuscript: “Here, photonic quantum emitter coupled systems relying on high-Q cavities, give rise to large emission enhancement, good modal coupling and guiding. However, storing photons for many cycles fundamentally limits emission rate and bandwidth. Plasmonic antenna’s such as plasmonic nanocubes^{3,46} enable emission enhancements of $>10^3$ and large cavity emission rates. However, the guided extraction of light in an antenna coupled quantum emitter system is difficult, hindering integration into PICs. Meanwhile, the bandwidth is limited by the antenna resonance and the spatial overlap of excitation and emission introduces challenges for excitation beam filtering. The integrated waveguide presented here in contrast, is non-resonant and hence inherently broadband, not bandwidth-delay product limited, separates excitation and emission spatially and provides a Purcell enhancement at par with or larger than photonic^{7,47} and plasmonic⁴⁸ waveguides, respectively.”

Reviewer #1 writes: 2. Figure 1c plots waveguide vs. plasmonic cavity Purcell factor as a function of wavelength. As the author already mentioned, the Purcell factor of cavity is determined by the Q and mode volume. Are the parameters used to plot Figure 1c representative? A fair comparison between resonant and waveguide structures, not limited to plasmonics, should be presented.

Our Response:

Yes, the parameters are representative. The plasmonic cavity case, $Q \sim 10$, is a representative value (Palstra et al., 9, 1513-1531, (2019)). The plasmonic waveguide $F_p = 3c\lambda^2 / (4\pi n^2 v_g A_m)$ curve is based on simulations where all wavelength dependent values were extracted for the presented waveguide structure. We have now added ref. [72] to the caption of Fig 1b to reflect that.

A general comparison of possible Purcell enhancements for different device geometries of photonics vs. plasmonic systems strongly depends on the specific parameters such as gap size, exact geometry, materials used etc. Hence a general comparison is out of the scope of our manuscript. However, following the valuable point raised by the reviewer, a better/fairer comparison than previously presented would be to normalise all presented enhancements to unity and emphasize only the enhancement bandwidth for the different systems. Indeed, this was our main intention for Fig. 1c, since we characterise the Purcell factor in depth at a later stage of the manuscript. We have now i) normalised all Purcell values to unity at the peak of the Erbium emission spectrum and ii) added the spectrum of a representative high Q photonic cavity to Fig 1c. The changes are also reflected in the figure caption.

To address the reviewer's question regarding the comparison to photonic waveguides: a general comparison of plasmonic with photonic waveguides, in contrast to cavities, is more intricate. Photonic waveguides with relation to the presented work are practical either in the form of a photonic crystal waveguide, a standard ridge waveguide or a slot waveguide configuration.

Photonic crystal slow light/ cavities waveguides:

In a slow light waveguide, the coupling bandwidth can be tens of nanometres at moderate Purcell enhancement factors (e.g. Laucht et al., PHYSICAL REVIEW X 2, 011014 (2012)). A radiative enhancement of about 55 has been reported. This approach is inherently bandwidth-delay product limited and only moderate Purcell enhancements are possible. Another approach is coupling a waveguide and a conventional cavity. Patterned photonic crystal waveguide cavities allow for large Purcell enhancements, however only at very limited bandwidth, due to the resonant mechanism. Reports related to Erbium coupling demonstrated enhancements on the order of 200 (Dibos et al, Nano Lett. 2022, 22, 6530–6536) to 700 (Raha et al., Nat. Comms., 2020, 11, 1-6) but at a linewidth \ll

1 nm. These leading results are only moderately higher than that reported in our manuscript across a considerable broader bandwidth.

Photonic ridge waveguides:

Simple ridge waveguides neither allow for strong modal confinement nor provide resonant enhancement (Q-factor). Hence, they are not suitable as an emission enhancing platform. Depending on the material they can provide comparably broadband coupling (see e.g. Errando-Herranz et al. ACS Photonics 2021, 8, 1069–1076). Ridge waveguides can allow for low-medium Purcell enhancement when patterned to a photonic waveguide cavity (Peyskens et al., Nat. Comms., 2019, 10, 4435), which however inevitably lowers the coupling bandwidth.

Photonic slot waveguides:

Photonic slot waveguides differ from plasmonic slot waveguides in at least two important aspects, namely i) larger mode area and ii) larger variation of the mode area dependence with wavelength. The first difference leads to limited Purcell enhancement factors. Here, Purcell factors lie in the range of 8 – 35 (Skljarow et al., PHYSICAL REVIEW RESEARCH 4, 023073 (2022)) reported at a similar wavelength range (C-band) as in our manuscript. The second difference leads to a strong dependency of the achievable Purcell enhancement as well as coupling efficiency to the fundamental slot mode on wavelength.

A general comparison to the different classes of photonic waveguides is out of scope of Fig 1c. However, we believe that the improved version of Fig 1c based on the reviewer's comment now depicts a fair comparison and illustrates the broadband nature of plasmonic waveguides in comparison to other commonly used structures.

We have now added a brief comparison and new literature to the manuscript to further contextualise and compare the investigated structures to the various waveguides structure in line 90 – 96:

“The plasmonic waveguide presented, provides a Purcell factor of >338 over a large bandwidth. In comparison, waveguide-coupled photonic cavities have been shown to provide Purcell factors of >700, but over sub-nm bandwidths³⁸. Although slow-light photonic crystal waveguides can extend bandwidth to a few nms, Purcell factors (~10)³⁹ are set by their delay-bandwidth product⁴⁰. Finally, photonic slot waveguides have been shown to provide modest Purcell enhancements in the range of 8 - 35⁴¹.”

Reviewer #1 writes: *3. In Figure S5 b, the two simulated coupling efficiencies for 60nm gap width are obviously different. Please clarify.*

Our Response:

It may be that the legend of Fig. S5b might be slightly misleading as it is not surrounded by a box which may have led to the false impression of an additional data point at 60 nm. To clarify this to the readers, we have now placed a box around the legend.

Reviewer #2 writes: *The manuscript by Gusken et al. demonstrates coupling of erbium ions to an engineered plasmonic nanowaveguide, showing Purcell radiative rate enhancements of ~ 300 for ensembles measured at room temperature. The plasmonic geometry includes both a gap nanowaveguide and a tapered nanofocusing region that allows efficient coupling of the former with a silicon slab waveguide located underneath the metallic region. I find the geometry promising, considering it can in principle provide relatively large Purcell enhancements alongside efficient transition into low-loss photonic waveguides.*

Our Response:

We would like to thank the reviewer for these insightful comments on which basis we were able to improve the manuscript. The rigorous and detailed feedback of the reviewer provided an important basis to further clarify the presentation of our experimental data and helped to increase accessibility and appeal to a specialised as well as the broader readership alike. We have addressed all questions and comments raised by the reviewer as follows. (Replies in green; Added parts in the manuscript in blue):

Reviewer #2 writes: *The particular geometry demonstrated here is novel regarding quantum emitter coupling (it has been originally developed by the authors and reported in prior publications). However, experiments by other groups (e.g. Dibos et al. PRL 120 243601 (2018)) have already demonstrated larger or comparable Purcell enhancement factors as observed here, using photonic structures. The main advantage that is claimed in the paper - broadband operation - is compelling, however somewhat limited in appeal. For instance, slow light photonic crystals (as discussed in Ref. [1]), and dielectric bowties such as in ACS Photonics 9 1647 2016) offer similar advantages and have shortcomings that are arguably at the same level as those of the present geometry. I think more context could have been provided to clarify the appeal of the work, particularly, but not limited to, regarding applications in quantum networks (for instance, which applications or measurements more specifically could such performance enable, in view of what has been demonstrated?*

Reviewer #2 writes: *Also, are there other unique characteristics, besides broadband operation, that could enabling?).*

Our Response:

There are indeed unique enabling characteristics, but the broad band operation should not be overlooked; very few systems have the broadband capabilities of our system. First, we would like to emphasize that the mentioned work by the reviewer (as presented e.g. in Dibos et al. PRL 120 243601 (2018) or ACS Photonics 9 1647 2016) is a photonic crystal cavity, which will naturally have a band limited enhancement. While the review makes this

clear already, we point out that the quoted enhancement by Dibos et al is only over a limited bandwidth, enhancing just a small portion of the Erbium luminescence. Photonic high-Q cavities provide their large emission enhancements by storing photons for many round-trips (large Q-factor), which inherently limits the rate at which photons can be released from the cavity (see e.g. Science 356 (6344), 1260-1264). Plasmonic waveguides in contrast offer a broadband mode continuum and allow for simultaneous broadband emission enhancement and fast emission rates. One of the core challenges of the plasmonic approach however is how to efficiently couple to, extract and guide light from a tightly confined plasmonic mode. We addressed this issue within our study and via different routes in another recent work (see Fu et al. Nature Nano., 17, 1251-1257, 2022).

With regard to the alternative broad band approaches mentioned by the reviewer: i) slow-light photonic crystals are inherently bandwidth limited to a narrow frequency range set by the bandwidth-delay product, as explained in our introduction. Such structures would not be able to provide enhancement across the full Erbium band, as shown in our work, without increasing the light velocity and thus reducing the Purcell effect. In this context, photonic crystal waveguides are often considered to be broadband when providing a Purcell enhancement spanning over a range of several nanometres (ACS Photon. 7, 2343–2349 (2020)). In our device we experimentally show an enhancement over tens of nanometres, the entire Erbium emission band, and theoretically hundreds of nanometres. We even suspect that the radiative emission pathways at 980 nm and 2.75 μm are also both enhanced, although this cannot be verified with our current system. The reviewer also suggests ii) Bow-Tie antennas. These will have large Purcell enhancements from strong confinement, but are still limited by virtue of being a resonant structure. Some residual confinement will be available within the gap region of such structures, but emission extraction would depend on the antennas scattering characteristics.

We would like to further elaborate on other enabling advantages of our device. Unique characteristics include:

- i) Plasmonic components are unique as they are the only technology which allows for simultaneous optical and electrical interaction at the emitter scale. As the waveguides are metallic, this opens up an entire field of opto-electrical emitter tuning right at the source i.e., at atomic scale. For example, the stark splitting can be further exploited for electrical frequency shifting of the enhanced quantum emission which is then coupled to the waveguide.
- ii) the facile integration with existing photonic integrated circuits. Here, the hybrid coupling approach presented is in particular interesting as it does not rely on tapering of the underlying photonic waveguide to feature sizes below critical values (e.g., 120 nm features) which are currently commercially available by e.g., DUV lithography in commercial Si foundries. This allows seamless integration into existing PIC fabrication processes within e.g., a post-processing top patterning step.
- iii) As described in the introduction (line 50,51 and 63), the non-resonant (i.e., no-Q) nature of the waveguide enables rapid out-coupling of photons. Hence, many

photons can be emitted potentially “on-demand” from the right quantum emitter. This is of high interest as an enabling interface or “launch pad” for single photon sources in quantum photonic networks when e.g., combined with in-plane excitonic dipole emitters in 2D materials. Here, the computational capability of such network scales with the amount of available coherent single photons. This number however is restricted in high-Q systems as the emission rate scales inversely proportional to the Q-factor. Rapid outcoupling of photons additionally bears high promise for room temperature single photon sources as fast emission rates allow to “out-pace” the phonon decoherence channels (ref. [13]).

- iv) The footprint of the structure is extremely small allowing for denser integration on costly SOI, which increases the amount of achievable circuit complexity and hence the photonic chip capabilities.

Light matter interaction at the emitter scale is a broad field where our structure allows to measure and explore simultaneous electrical and optical interaction at the nanoscale which is of interest to develop a better fundamental understanding of this regime. With respect to applications, we would like to highlight the very promising new approach the plasmonic waveguides bring to the challenge of integrating single photon sources for quantum integrated circuits (qPICs).

In order to emphasize the unique characteristics and its potential in the field of single photon source integration on qPICs, we have now added the following sentences in line 97 – 104:

“The non-resonant hybrid waveguide device combines several unique characteristics: i) broadband emission enhancement, ii) rapid, efficient and guided photon extraction from quantum emitters, iii) facile and compact integration into photonic networks and iv) the possibility to simultaneously address emitters optically and electrically to e.g. tune Stark split states by external electrical fields⁴². The device – when combined with a hybrid integration approach⁴³ – provides a novel, scalable route with the potential to enable next-generation on-demand⁴⁴ single photon sources for quantum photonic integrated circuits (qPICs)⁴⁵.”

Reviewer #2 writes: *The methods are reasonable in principle, however further clarification is needed as indicated in my comments below. The text needs some work as well for clarity. For these reasons I feel I am unable to recommend publication in Nature Communications. It might be acceptable if my concerns above about the general appeal are properly addressed, as well as my comments below.*

Our Response:

We thank the review for their thorough review of our manuscript. We have addressed all question and comments raised by the reviewer as follows.

Reviewer #2 writes: 1 - Fig. 1a is somewhat misleading because the fabricated devices do not feature an etched photonic waveguide as depicted, but rather a silicon slab. I think the fact that a different geometry was tested should be better clarified in the text, including in the Figure 1 caption. I also think that the reason for building the geometry in Fig.1a should be expanded in the introduction, as well as the reasoning for utilizing the slab, rather than the waveguide geometry, in experiments.

Our Response:

We thank the reviewer for pointing this out. We agree with the reviewer that the inset of Fig 1a is not the exact device under investigation and can be misleading.

In order to avoid additional fabrication steps (such as waveguide etching, oxide filling and planarization), we coupled the emission from the plasmonic slot mode to a guided hybrid Si slab waveguide mode (shown in Fig 1a, b, h) and not to a Si ridge waveguide mode (shown in the inset of Fig 1a) prior coupling to the grating coupler coupling. The patterning of the photonic waveguide would have added significant fabrication expenses which are not needed at this stage and do not change any part of the study or the results. The typical width of a standard SOI ridge waveguide in the C-band is 450 nm – 600 nm. Hence, compared to the plasmonic gap sizes of 10 – 60 nm it effectively is very similar to a slab. The device presented, using a slab waveguide, is a viable alternative to the Si ridge waveguide coupled device but does not require Si patterning while demonstrating the same underlying physics. We believe however that it is useful to illustrate both geometries at least conceptually to put our device into context for e.g. quantum photonic networks and the PIC and integrated single photon sources community in general for which patterned waveguide coupling might be more applicable.

In order to clarify the difference and to clearly point out that the inset in Fig 1a is merely an illustration of the practical application example of this device in a photonic integrated circuit (PIC), we now made the following changes:

- i) We adapted the inset illustration of Fig1a as well as Fig 1b to make the dimension relations more to scale.
- ii) We adapted the figure caption accordingly.
- iii) We added a sentence to the introduction in lines 71 - 73 *“This is realized by stacking a tapered plasmonic waveguide on top of an oxide buried photonic slab waveguide. While this keeps fabrication to a minimum, coupling to buried ridge waveguides is equally possible.”* to point out that a Si Wg instead of a Si slab can be used for direct PIC integration.

Reviewer #2 writes: Lastly, what is the simulated dipole coupling efficiency into the ridge silicon waveguide? Is it as good as the claimed coupling into the silicon slab mode?

Our Response:

As explained in our answer to the previous question, we have not considered coupling between the slab waveguide and a ridge waveguide. We anticipate that the coupling would be quite reasonable, with appropriate engineering. For a consideration of dipole to waveguide coupling efficiencies, see our response to a later comment by this reviewer.

Reviewer #2 writes: *2 - On line 118, it is claimed that the emission into the plasmonic gap mode dominates over nonradiative pathways. I think the authors should expand this discussion, with some quantification. What nonradiative pathways can be reasonably expected, and what are their time-scales? The transitions that are targeted are fairly slow (\sim ms), and the enhancements that are expected only bring them to the \sim us scale. Are these time-scales sufficient?*

Our Response:

Erbium luminescence is known to be radiatively efficient, implying an intrinsically slow non radiative lifetime compared to the radiative one. By enhancing emission rates to the us range, these intrinsic non-radiative processes are even less relevant to the erbium ion efficiency. We are more concerned with extrinsic non-radiative process introduced because of the optical confinement. However, we show and discuss in Fig. 2 c,e,f) and (d), the observed lifetime reduction is accompanied by a commensurate enhancement in photon collection. Non-radiative pathways, such as quenching of ions too close to the metal, would result in a decrease of both, lifetime, and photon collection. This suggests that although non-radiative processes might be present, they are not dominant.

We also discuss (please see line 192 to 207) and quantify (please see Fig 3) the radiative enhancement i.e. the radiative channel, as part of the total lifetime reduction. Here, the Purcell factor, i.e. the radiative rate's enhancement factor, is explicitly determined from PL saturation measurements, which disentangles this radiative part from the total emission, which includes both non-radiative and radiative contributions. Based on this, we were also able to show that the radiative channel is the predominant channel.

In general, an emitter in the vicinity of a metal surface can be subject to emission quenching, which is a non-radiative process. This results in a decreased lifetime. Its origin can be e.g. Joule heating or coupling to nonradiative higher order plasmonic modes [please see ref. [50]]. Quenching for an emitter-to-metal distance of >9 nm however is minor (please see ref. [50]). In metal-insulator-metal structures, the emission quenching for even smaller distances is also weak (see ref. [3] and ref. [51-53]). Here, as explained in ref. [53], coupling to higher order plasmonic modes is a predominant non-resonant pathway. However, fluorescence quenching of Erbium in SiO₂ close to Au has been experimentally reported to lie in the 1-2 ms range (Nanfang Yu et al 2009 New J. Phys. 11 015003).

We have now added this information to line 148: “Fluorescence quenching due to coupling to higher-order plasmonic modes⁵⁰, is expected to be minor^{50–52} with a slow non-radiative decay⁵³.”

Reviewer #2 writes: 3 - More experimental details should be given in Supplementary Section 3, regarding propagation length and coupling efficiency estimates. The text is somewhat confusing and the explanations not sufficiently clear. From the main text, I understood that propagation losses were obtained by measuring waveguide-outcoupled PL intensities for waveguides of varying lengths, for the same pump level. However, the SI text suggests that a passive cut-back transmission measurement was performed for such estimation, where a 1550 laser beam was launched into one grating and the transmitted beam power was measured.. Is that the case? The authors should make this more clear in the SI text.

Our Response:

We thank the reviewer for this comment. Yes, that is indeed the case. Propagation losses were obtained using the cut-back method based on a laser transmission measurements at Erbium emission wavelengths. Here, a laser beam was launched into one grating and the transmitted power was measured at the outcoupled grating, for various waveguide lengths. The experimental results were confirmed by 3D FDTD simulations and in previous experiments [see e.g. ref. [31] & ref. [22] Nielsen et al. Science (1979) 358, 1179–1181 (2017)].

We have now adjusted the sentence in line 180 of the manuscript: “[..] as extracted from transmission measurements...”.

We also adapted section 3 of the Supplementary information to make this clearer. We have added further information about the experimental details in line 86 ff. :

“To accurately estimate i) the propagation length of the hybrid plasmonic mode as well as ii) the coupling efficiency from light exciting the plasmonic gap to the silicon slab waveguide mode, a passive waveguide transmission measurement was performed^{7,8}. Here, a CW focused laser beam was coupled in via one grating as e.g. shown in Figure S2b and coupled out via the opposite grating after propagating through the plasmonic gap structure. The input and output light was incident and collected via the same objective, respectively. However, an iris positioned in the image plane after the objective (c.f. Figure S13) allowed to isolate the signal from the out-coupling grating from the rest of the sample. The input beam was incident under a slight angle created by coming in off-axis through the objective. The in ($R_{cts,in}$) and output ($S_{cts,out}$) signals have been measured (corrected for the background signal as explained below) for various devices of different gap widths in dependence on waveguide length. Based on this information, the system’s coupling and guiding efficiency comprising (grating in-coupling, Si-waveguide to plasmonic waveguide, plasmonic propagation, plasmonic waveguide to Si-waveguide and grating outcoupling

efficiency) was estimated in dependence on the plasmonic waveguide gap length l . This is reflected in $\eta_{meas}(l)$ i.e., Eq. S3.1 and Eq. S3.2 and shown in Figure S4 and Figure S5.”

Reviewer #2 writes: *Importantly, though, the coupling efficiencies $\eta_{\{inc,outc\},gap}$ in this case only correspond to the coupling efficiency between gap waveguide mode and the silicon slab. It does not correspond to the coupling efficiency of the Er emitters into the slab, because, strictly, the emitter may couple to radiative (and also possibly bound) modes, in addition to the gap mode.*

Our Response:

$\eta_{\{inc,outc\},gap}$ is the coupling efficiency from the plasmonic waveguide mode to free space. In our experiments, we compare collected emission from plasmonic devices with a control device. Since identical gratings are used in both types of device, the reviewer is essentially correct: we assess the ratio of efficiencies, $\epsilon_{(c,Gr)}/\epsilon_{(c,Gap)}$, which measures the coupling efficiency from the plasmonic waveguide mode to the slab waveguide. The coupling efficiencies of the gratings is normalised out.

Erbium emission may couple to a variety of modes of the slab and also modes that radiate into the substrate and air. However, we re-iterate, that the enhancement in emission rate and photon collection rate are similar in all devices. This indicates that the enhanced erbium emission is dominantly being collected by the out-coupling gratings. See response to next comment for more details.

Reviewer #2 writes: *The authors should provide simulation results quantifying the expected dipolar coupling efficiency to the plasmonic gap mode, and/or to the slab waveguide after the taper. This is mostly to rule out, theoretically, the possibility that coupling to unbound modes is significant and possibly comparable the coupling rate to the (bound) waveguide mode. This coupling efficiency, the so-called modal β -factor, is very important, because it indicates how ultimately efficient the light-matter interface can be. Importantly, all measurements done to characterize such coupling only takes into account upwards emitted light, disregarding dipole emission that exits through the substrate. The waveguide-coupled light, then, is not the total that is emitted (though might be a good portion of it).*

Our Response:

This is a very interesting question, and one that we had not investigated in great detail. We have now performed additional 3D FDTD simulations to estimate the dipole-to-waveguide coupling efficiency, i.e. the beta factor, resulting in a $\beta \sim 80\%$. This is consistent with other literature (PHYSICAL REVIEW B 78, 153111 (2008)). An unforeseen loss channel is coupling to in-plane waves propagating in the gap between the Au waveguide and the Si slab, accounting for $\sim 14\%$ of loss. The relatively high proportion is likely due to the available

confinement and also the 2D continuum of the in-plane radiation modes. We note that in future designs, this could be eliminated by using parallel gold wires to define the gap region, or by an interrupted Si-waveguide underneath the plasmonic gap section. This would eliminate the continuum of 2D modes. Other experimental work shows that high β -factors are possible for Raman scattering in such a parallel wire gap structure (see Fu et al. Nature Nano., 17, 1251-1257, 2022).

We have also computed the proportions of light radiated into the substrate which account for $\sim 1\%$ of the emission. Of course, the gap structure enhances waves moving both in the plane (1D waveguide mode) and out of plane (2D slot mode). The gap thus enhances the radiation rate of light normal to the device surface, but this is a small contribution here. The computation also considers the effect of quenching (i.e. coupling to plasmonic higher order modes which do not necessarily contribute to the far-field emission), which might account for the additional loss. We are glad to include these new numerical data and we will investigate these finding more deeply in the future.

We have now added the following section to the Supplementary Information S18, line 498 - 510:

“This lower bound estimate assumes a unity coupling efficiency from the dipole emission implanted below the plasmonic gap waveguide to the gap waveguide mode. This coupling efficiency can be quantified by the well-known β -factor, shown over the studied wavelength range in Figure S11.

Here we show dipole emission coupling to the gap mode of a 10 nm wide plasmonic waveguide. The emission coupling with emitters implanted in the SiO₂ spacer layer right below the gap (as in the experiment) shows a β -factor of $\sim 80\%$. Here, emission channels other than the plasmonic waveguide are coupling to an in-plane mode at the oxide interlayer at the interface of the Au waveguide and the Si slab ($\sim 14\%$ of loss), as well as quenching (i.e. excitation of higher order plasmonic modes). Light emitted into the substrate accounts for $\sim 1\%$ while upward radiation is minor (see above). For comparison we also show the case for an emitter implanted within the gap. The coupling efficiency can potentially be further increase by choosing a different refractive index within the gap

(i.e. filling it with a higher index polymer). While the results are well in line with simulations results from the literature²⁶, experimental works show that even β -factors near unity are possible in plasmonic gap waveguides²⁷.”

Reviewer #2 writes: 4 - *Have the authors tried to perform transmission measurements with the Er³⁺-doped gap waveguides? It would be very interesting to see if a significant enhancement of the single-ion extinction cross-section due to the gap mode can be observed. The extinction has a direct dependence on the β -factor (see e.g. Turschmann et al., Nanophotonics 8 1641 2019).*

Our Response:

This is indeed a very nice suggestion and certainly in plans for future work. However, this would require reaching lifetime limited linewidths as it is a coherent interference effect. As such this would require cryogenic cooling, which we have plans to do but unfortunately this is not in the scope of this investigation.

Reviewer #2 writes: 5 - *What accounts for the shorter lifetime observed from the grating device (~ 3 ms) in contrast with the lifetime of bare Er³⁺ ions implanted in SiO₂?*

Our Response:

Here, implantation density as well as the vicinity to the Si slab can play a role. [please see ref. [55, 56]]. However, we would like to emphasize that all samples are constructed on the same SOI substrate, and as such have identical Erbium implantation density and depth. They only differ with respect to the plasmonic device structure used, allowing for a direct comparison with control samples to identify the impact of the plasmonic gap waveguide.

We have now added a sentence in line 214 to point this out: “Here, the presence of a Si slab⁵⁵ and high doping concentrations⁵⁶ can explain such a lifetime reduction. Nonetheless,..”

Reviewer #2 writes: 6 - *Can the authors further clarify the difference between $\tau_{Gr, \lambda}$ and τ_{Gr} , and similar for the gap devices? Two enhancement values, $\tau_{Ggap, \lambda} / \tau_{Gr, \lambda}$, and τ_{Ggap} / τ_{Gr} are given in the text (lines 162 and 188 respectively), and I don't understand exactly why. As far as I can tell, they are obtained for the same device, under the same conditions, and are enhancement averages over all probed wavelengths.*

Our Response:

Thank you for the question. The difference between the two lies in the analysis method and how the values are extracted from the measurement.

τ_{Gr}/τ_{Gap} is the lifetime ratio obtained from averaging across the entire spectrum for different acquisition powers. The according emission rates are shown in Fig 2f in dependence of power from which the lifetimes/rates at zero power can be extracted via extrapolation.

$\tau_{Gr,\lambda}/\tau_{Gap,\lambda}$ is a spectrally resolved lifetime ratio i.e. the lifetime for each wavelength extracted from extrapolation of the lifetime (or decay rate) to zero pump power. The result is shown in Fig 2 c, e. $\tau_{Gr,\lambda}$ and $\tau_{Gap,\lambda}$ are extracted in a similar measurement as the one shown in Fig 2f but per wavelength and not averaged over the entire spectrum.

In general, density effects and a pump power dependence of the lifetime can play a role in lifetime extraction. Here however, both values were extracted from the measurements by extrapolation to the zero-pump-power value, as explicitly shown for the averaged lifetime ratio across the spectrum in Fig 2f. The values extracted from both methods agree well with each other.

We have now adapted the captions of Fig 2 c,e and f accordingly.

We have furthermore rephrased lines 217 ff.: " $\tau_{(Gap,\lambda)}$ was determined by extrapolating the lifetime measured at each wavelength and for various powers to zero power excitation, excluding power dependent effects. Additionally, Figure 2f provides the emission rates, $\Gamma_{Gap} = \tau_{Gap}^{-1}$, for the waveguide and, $\Gamma_{Gr} = \tau_{Gr}^{-1}$, for the grating device. These were averaged across the spectrum for each excitation power. [...]. The mean of the spectrally resolved emission rate agrees well with value extracted from the spectrally averaging." to highlight the difference between the two.

Reviewer #2 writes: 7 - *What accounts for the linear increase of the emission rates with the pump power in Fig. 2f? For what pump powers were $\tau_{Ggap,\lambda}/\tau_{Gr,\lambda}$, and/or τ_{Ggap}/τ_{Gr} obtained?*

Our Response:

This comes from the saturation of the Erbium ion emission and up-conversion which both depend on pump power [see e.g., ref. [56]]. In order to determine the lifetime, the experiments must be conducted at low intensity. This cannot be done easily due to the relatively low count rates, so we must use linear interpolation of the power dependent lifetime data. Here, all lifetimes were extracted in the limit of low pump power – i.e. extrapolated to zero power.

Reviewer #2 writes: 8 - In the caption of Fig. 3 there is a comment that pump powers were limited to avoid changes to device morphology. Can the authors expand on what takes place and why?

Our Response:

In general, high powers can damage the gold nanogap waveguide due to heating. However, from previous calibration measurements we knew that 40 mW would be a safe upper power level to operate at on the final devices. More importantly, the devices used during the experiment did not show any sign of damage or morphology change within the range of pump powers used, i.e. they were not tested to destruction.

We have reformulated this phrase. It now reads, “Note that the samples were not subject to intensities that could cause any changes in morphology (i.e., damage).” in the caption of Fig. 3.

Reviewer #2 writes: 9 - The difference in gap waveguide in/out coupling (quantified by $\eta_{inc,gap}$ and $\eta_{out,gap}$) is somewhat curious, I think the authors should expand on this somewhat. Since the structure is completely symmetric, I would expect in- and out-coupling to be reciprocal, unless there are additional input or output modes that are somehow involved. Can the authors provide more details about how the simulations to calculate this were performed?

Our Response:

In our cut back method experiments, we can compute the coupling efficiency across the entire structure. We can extract an experimental value of $\eta_{inc,gap} \times \eta_{out,gap} \times \eta_{grating}^2$. To determine the separate values of $\eta_{inc,gap}$ and $\eta_{out,gap}$, we use the experimental values for $\eta_{grating}$ and a numerical value for $\eta_{out,gap} = 0.88$. This numerical value was determined by launching the fundamental eigenmode of the plasmonic waveguide, using a mode source in 3D FDTD simulations, and measuring its out-coupled power into the fundamental mode of the Si slab. With this and the expression stated above one arrives at a value of $\eta_{inc,gap} = 0.8$. We note that if we had assumed equal in and out coupling, we would have found values of $\eta_{out,gap} = \eta_{inc,gap} = 0.84$.

The asymmetry here is necessary to include to account for the ability of the grating to form converging waves on the plasmonic gap. The curved grating couplers create circular waves of the in plane coupled light enabling high but not perfect coupling into the fundamental hybrid mode (about $\sim 80\%$). However, on de-focussing, the plasmonic gap modes simply spreads out into waves of the slab that are all scattered by the output grating. So, the field distribution in the slab at the input grating is not the same as the field at the output grating.

In other words, in-coupling, i.e. slab-to-gap coupling can be less efficient as for the opposite way (i.e. gap-to-slab) if not all light is perfectly coupled to the focused slab mode priorly.

As described in the SI line 110 ff., the in-coupling of light launched via grating couplers into the fundamental mode is more prone to scattering into other modes than coupling from the confined gap waveguide mode out and into the fundamental hybridized slab mode (i.e. in the opposite direction).

We have now added the following sentence to the Supplementary information, line 137 - 140:

“The asymmetry in values of $\eta_{inc,gap}$ and $\eta_{out,gap}$, is necessary to account for the imperfect focussing of waves in the slab by the curved gratings. The focussed field at the gap does not perfectly match the field distribution of the gap mode. However, on de-focussing, the plasmonic gap mode spreads out into waves of the slab that are all scattered by the output grating.”

Reviewer #2 writes: 10 - *In Fig. 1a, why does the off-gap spectrum not display the same characteristic peak as the other curves? Is the spectrum just due to the SOI background emission, as in Fig. S6?*

Our Response:

We assume that the reviewer is referring to Fig 3b and not Fig 1a.

The off-gap spectrum does not show the distinct peak of the other devices as most emission couples in-plane to the Si slab waveguide as no gratings are present to project the k-vector out of plane. The spectrum in fact is very similar to the Gr reference sample spectrum except the Erbium emission peak around 1536 nm only really becomes visible when coupled out of the sample and into free space via a grating.

The SOI background was measured on separate control samples with metal structures, but without implanted erbium ions. The background does overlap with the ion emission, however, as shown in FigS6, it is about 50% of the signal compared to the “off-gap” spectrum. The majority of the “off-gap” signal is due to Erbium emission.

We have now mentioned this explicitly in the caption of Fig 3b in the manuscript: “Note that the common Si-background has not been subtracted at this point. It contributes about half of the “off-gap” signal (Fig. S6).”

Reviewer #2 writes: 11 - *In Fig. S1, normal excitation pump leads to a standing-wave in the gap. Why is that? Does this mean that the excited ions are only those located within the hotspots? Is this taken into account in the gap ion contribution ($S_{\backslash Gap(w)}$) estimation?*

Our Response:

During the out-coupling of light from the gap waveguide and into the slab, small back reflections occur at both ends of the waveguide (please also see SI 1 line 40 ff.). This leads to the interference of light in the gap, which is launched due to scattering at the waveguide edge. However, this is a weak effect and not easy to model as it depends on the nanoscale curvature of the taper where the gap is most narrow. As verified by estimations based on the interference pattern, one can expect a reflection of about 6% from the ends of the waveguide. We have now added this estimate of the reflection using the interference visibility in section SI 1 of the supplementary information, line 50-58: “Please note that the interference pattern observed in Figure S1 a1 stems from weak back reflections at both ends of the waveguide induced by 980 nm excitation light scattered into the waveguide mode. The reflectivity can be estimated to be about 6% based on the intensity distribution of the simulated gap field in Figure S1 a1. Here, $I = |a_{gap}e^{ikz} + a_{gap}re^{(-ikz)}|^2$, with the amplitude, a_{gap} , the reflection coefficient r and propagation constant k . Meanwhile the intensity maxima and minima from Figure S1 a1, $I_{max} \approx 3.3$ and $I_{min} \approx 1.2$ yield the average signal intensity of 2.25. Re-writing the intensity expression to $I_{\pm} = (a_{gap})^2(1 + r^2) \pm 2(a_{gap})^2r$, then the average intensity $(a_{gap})^2(1 + r^2) = 2.25$. Re-writing and eliminating a yields $1.05/2r(1 + r^2) = 2.25$ from which we can estimate the reflectivity $r^2 \approx 6\%$.”

Please also note that the field amplitude of the field depicted in Fig S1 at the pump wavelength of 980 nm only shows very moderate enhancement. As described in line 43 in the SI, as well as in the main text, an average field enhancement of about 2 can be observed.

Regarding the second question: the variation in pump intensity in the gap is not taken into account in our model. Since the modulation is weak, the pump excitation averages out over the length of gap region.

Reviewer #2 writes: 12 - On Fig. 3c, the luminescence intensity curves show signs of saturation at increasing powers, however a clear saturation level is not apparent. Because of this, it is difficult to know at what level the ion populations in the waveguide and reference (no-waveguide) samples are being pumped in their respective saturation curves. I think the authors should estimate, based on the Fig. 3c data, at what level of saturation the populations in the waveguide and reference samples are, for the 40 mW power level considered for extracting EF.

Our Response:

Yes, we extracted the saturation parameters from the fitting model (S6.8) derived in the SI 6 by fitting it to the experimental data shown in Fig. 3c. Please find the resulting parameters in Table S1.

However, in order to provide a lower bound estimate which does not rely on any assumptions made in our model, we used the measured intensity values for gap and grating

samples at a pump power of 40 mW. This is legitimate since the ratio of the integrated luminescence signals monotonically increases with pump power.

Reviewer #2 writes: *13 - The authors should expand the discussion of why the presumed Stark-split peaks are visible in the present experiments due to enhancement of ED transitions. In particular, the role of MD transitions in masking ED transitions in non-enhanced samples would be better clarified - are the two types of transitions expected to overlap spectrally in the SiO₂ host, and have the feature comparable transition probabilities?*

Our Response:

Thank you for the comment. Yes, the MD and ED transition of Er³⁺ overlap spectrally and have very similar transition probabilities [please see: PHYSICAL REVIEW B 89, 161409(R) (2014); Nano Lett. 2016, 16, 5191–5196]. This leads to a fairly broad Er emission band near 1530 nm, as shown in our data. In a future study, Europium instead of Erbium could be used to distinguish MD from ED transitions as they do not spectrally overlap in Europium. We have now expanded our initial discussion and explicitly mentioned the overlap of MD and ED transitions. Please see in manuscript the new sentence; “Meanwhile, the electric dipole (ED) and magnetic dipole (MD) transitions of Er³⁺-ions overlap spectrally and are of similar magnitude^{64,65}.” in line 302.

Reviewer #3 writes: *In this manuscript, the authors described emission enhancement by using a plasmonic nanowaveguide based on Purcell effect. The authors selected Er-ion doped platform at telecommunication wavelengths, which is of broad interest to the photonic community. The enhancement effect is convincingly measured and presented, and the authors built a theory to explain some of the behaviors. Overall, this is a paper of high quality, and the topic is interested to the broad audience of photonics. I recommend it for publication in Nature Communications.*

Our Response:

We thank the reviewer very much for the positive feedback and the appreciation of the details investigated in this study. It is indeed encouraging to read that our work found appeal for an expert audience. We further want to thank the reviewer also for the comments which helped to improve the manuscript further. We address those comments in the following (Replies in green; Added parts in the manuscript in blue):

Reviewer #3 writes: *I only have two minor comments:*

(1) In figure 2, it is difficult to understand why the large enhancement effect on lifetime in panel (c) only translates to a few times enhancement of luminescence in panel (d). Could the authors provide a simple, intuitive explanation for this?

Our Response:

Thank you for this question. In Fig 2d we show a comparison of raw spectral data from the various devices considered. (i.e. Wg vs Gr). These data are now normalized by the number of excited ions. Please note that ions are homogeneously implanted across the sample, but the metalized regions of the gap devices greatly reduce the number of ions that can be excited. The raw data was shown to illustrate the already large enhancement which can be directly observed without taking any normalization factors into account. However, of course, this is a serious underestimate of the real enhancement as merely a fraction of ions is excited in the Wg device in comparison to the Gr device. The gold regions in the waveguide device mask the pump light to excite < 12% of the ions that contribute to the grating device measurement. We also show that while only 0.3% of all ions sit in the gap, they provide 80% of the emission.

We also note that the lifetime measurements, shown in Fig 2c, are independent of the number of ions and only dependent on the environment they are embedded in. The panels together show that both, emission enhancement and lifetime reduction occur due to the plasmonic waveguide device.

We have now added the sentence, "Note that the spectra have not been normalized by the amount of illuminated ions. In fact, For the Wg device only a small fraction of the number of ions excited in the Gr device contribute to the signal", to the caption of Figure 2 to be

clearer about this point. Additionally, we would kindly refer the reviewer to lines 200 – 207 for a more detailed description.

Reviewer #3 writes: (2) *In figure 2 (a), the structure is labeled as “Wg,” but in panel (c) and (e), the authors use “gap” to label results for the waveguide. This is a little bit confusing. I feel it’s better to use “wg” everywhere throughout the manuscript.*

Our Response:

Thank you for this comment – we apologise for the confusion here. We have labelled the plasmonic device as “Wg”. Fig 2 (d) shows total collected emission, which includes emission from ions in the gap and from the taper regions, outside of it. These subsets of ions are exposed to different enhancements. We use the Wg label in Fig. 2 (d) since we are referring to all collected emission from all ions. However, in Figs. 2 (c), (e) and (f) the subscript “Gap” indicates that we are only considering ions in the *gap* of the device, which experience the peak enhancement. The lifetime measurement extracts the fastest component, corresponding to the ions in the gap. We have adapted the Fig. 2 caption to make this clear.

“Note that the Wg device contains ions excited in the plasmonic gap and in the taper region, which are subject to different rate enhancements.” And “[...] (including taper and gap region of the Wg device).”

REVIEWERS' COMMENTS

Reviewer #1 (Remarks to the Author):

The revised manuscript is well-organized and well-written. The characteristics of this device are summarized effectively, allowing readers to easily grasp the highlights of this work. A comprehensive comparison between this work and other devices has been provided to demonstrate the unique performance of this work. Based on the above reasons, I highly recommend this work for publication in Nature Communications. I believe this work will be interesting to a wide range of readers from the photonic communities.

Reviewer #2 (Remarks to the Author):

I find that authors have addressed my comments sufficiently well, so I am happy to recommend publication in Nature Communications. I have a suggestion to make the paper a bit more self-contained: please add a brief explanation about why the presence of a Si slab and a higher density of ions leads to faster decay times. Refs. 55 and 56 are cited, however no explicit explanations are given, which is somewhat unsatisfying. This is optional, though, and mostly for the readers' benefit.

Reviewer #3 (Remarks to the Author):

The authors have addressed all my concerns.

REVIEWERS' COMMENTS

Reviewer #1 (Remarks to the Author):

The revised manuscript is well-organized and well-written. The characteristics of this device are summarized effectively, allowing readers to easily grasp the highlights of this work. A comprehensive comparison between this work and other devices has been provided to demonstrate the unique performance of this work. Based on the above reasons, I highly recommend this work for publication in Nature Communications. I believe this work will be interesting to a wide range of readers from the photonic communities.

Our response:

We thank the reviewer for the positive feedback and especially the constructive and interesting questions which helped to improve the manuscript and increased its appeal for a broader readership.

Reviewer #2 (Remarks to the Author):

I find that authors have addressed my comments sufficiently well, so I am happy to recommend publication in Nature Communications. I have a suggestion to make the paper a bit more self-contained: please add a brief explanation about why the presence of a Si slab and a higher density of ions leads to faster decay times. Refs. 55 and 56 are cited, however no explicit explanations are given, which is somewhat unsatisfying. This is optional, though, and mostly for the readers' benefit.

Our response:

We thank the reviewer for this suggestion. This is indeed an interesting and valuable point which however would require a lengthier explanation as both effects are somewhat distinct. In order to avoid deviating from the key findings of the report, we hence would prefer to refer the reader to the provided references at this point. We believe this would benefit the description of the findings by keeping it as concise as possible.

At this point we would also like to thank the reviewer for the numerous well thought through questions which enabled us to re-think certain sections of the manuscript and take another perspective on our explanations. The reviewers feedback certainly improved the manuscript and raised interesting questions for potential future works.

Reviewer #3 (Remarks to the Author):

The authors have addressed all my concerns.

Our response:

We thank the reviewer for the constructive feedback during the review process. We furthermore welcome the reviewer's helpful questions which steered us to improving important points and how they are communicated within the manuscript.